# 🫐 BLEUBERI:
# BLEU is a surprisingly effective reward for instruction following

**Yapei Chang**
University of Maryland, College Park

**Yekyung Kim**
University of Maryland, College Park

**Michael Krumdick**
Kensho

**Amir Zadeh**
Lambda AI

**Chuan Li**
Lambda AI

**Chris Tanner**
Kensho

**Mohit Iyyer**
University of Maryland, College Park

## Abstract

Reward models are central to aligning LLMs with human preferences, but they are costly to train, requiring large-scale human-labeled preference data and powerful pretrained LLM backbones. Meanwhile, the increasing availability of high-quality synthetic instruction-following datasets raises the question: can simpler, reference-based metrics serve as viable alternatives to reward models during RL-based alignment? In this paper, we show first that BLEU, a basic string-matching metric, surprisingly matches strong reward models in agreement with human preferences on general instruction-following datasets. Based on this insight, we develop BLEU-BERI,[1] a method that first identifies challenging instructions and then applies Group Relative Policy Optimization (GRPO) using BLEU directly as the reward function. We demonstrate that BLEUBERI-trained models are competitive with models trained via reward model-guided RL across four challenging instruction-following benchmarks and three different base language models. A human evaluation further supports that the quality of BLEUBERI model outputs is on par with those from reward model-aligned models. Moreover, BLEUBERI models generate outputs that are more factually grounded than competing methods. Overall, we show that given access to high-quality reference outputs (easily obtained via existing instruction-following datasets or synthetic data generation), string matching-based metrics are cheap yet effective proxies for reward models during alignment. We release our code and data at `https://github.com/lilakk/BLEUBERI`.[2]

## 1 Introduction

Modern LLM alignment often relies on reinforcement learning with a *reward model* that guides the LLM to follow instructions according to human preferences [40]. Reward models are expensive to train, requiring large-scale human preference data and powerful backbone models [39, 55, 60]. Meanwhile, the emergence of high-quality instruction-following datasets (e.g., OpenHermes, Magpie, LIMA) has enabled cheaper *reference-based* alignment via supervised fine-tuning (SFT) [57, 73, 80].

---

[1] BLEUBERI stands for "**BLEU**-**b**as**e**d **r**eward for **i**nstruction following."
[2] Correspondence to `yapeic@umd.edu`.

39th Conference on Neural Information Processing Systems (NeurIPS 2025).

This contrast raises a natural question: **can we align language models with simple reference-based metrics in place of learned reward models?**

**Challenges with reference-based rewards:** On the surface, replacing reward models with reference-based metrics seems problematic. Obtaining high-quality references for complex, open-ended tasks can be even more expensive than collecting preference judgments, and many instructions (e.g., creative writing) lack a single ground-truth answer. Furthermore, aligning an LLM requires balancing multiple criteria (e.g., helpfulness, harmlessness, factuality) [4], that a single reference may not fully capture. Finally, automatic metrics that score LLM responses against references rely on unreliable and gameable methods like $n$-gram matching or embedding similarity, which has historically discouraged their use in LLM alignment.

**BLEU is surprisingly effective at modeling human preferences:** Despite these limitations, we find that BLEU [41], a simple string-matching metric long deemed inadequate for open-ended language generation [70, 10, 58, 44, 12, 33, 5], rivals large reward models in modeling human preferences. In experiments on general instruction-following tasks in the LMSYS chatbot_arena_conversations dataset, BLEU with five synthetic references achieves almost the same agreement (**74.2%**) with human preferences as a powerful 27B-parameter reward model (**75.6%**). We observe that reference quality is critical: references generated by powerful LLMs (e.g., Claude-3.7-Sonnet, GPT-4o) yield significantly higher agreement than those from weaker models. Our analysis reveals that BLEU's strong alignment signal comes from rewarding properties like factuality and proper formatting that are critical to instruction following tasks.

**BLEUBERI: directly using BLEU as a reward for RL-based alignment.** Motivated by the unexpectedly high agreement between BLEU and human preferences, we propose BLEUBERI, which uses RL to optimize BLEU on general instruction-following data. Previous efforts that used $n$-gram metrics like BLEU as rewards [43, 68, 16] faced obstacles such as unstable training [3, 45] and degraded output quality [30, 19, 21]. We revisit this line of work using modern LLMs within the paradigm of *reinforcement learning with verifiable rewards* (RLVR) [24, 62], which shows the effectiveness of simple, transparent rewards. BLEUBERI treats BLEU as a verifiable reward for general instruction-following tasks, using group relative policy optimization (GRPO) [50] to optimize a pretrained base LLM. We apply GRPO with BLEU rewards on a subset of challenging instructions for which the base model's outputs initially have low BLEU.

**Strong instruction-following performance without a reward model:** Across three different base models and four diverse instruction-following benchmarks, including ArenaHard[26] and WildBench[28], BLEUBERI matches (and sometimes exceeds) the performance of reward model-guided RL and SFT according to both automatic and human evaluations. This is a striking result, given the simplicity and cost-effectiveness of BLEUBERI compared to training and deploying large reward models for RLHF. Human evaluators find that BLEUBERI-trained models are just as good as those from models aligned with reward models. Furthermore, BLEUBERI models produce more factually-grounded responses than those aligned with either reward models or SFT. **Taken as a whole, ours is the first work to show that optimizing BLEU–far from overfitting to superficial $n$-gram matches–actually promotes helpful, factual, and well-formatted responses on general-domain instruction-following tasks.** Given access to high-quality

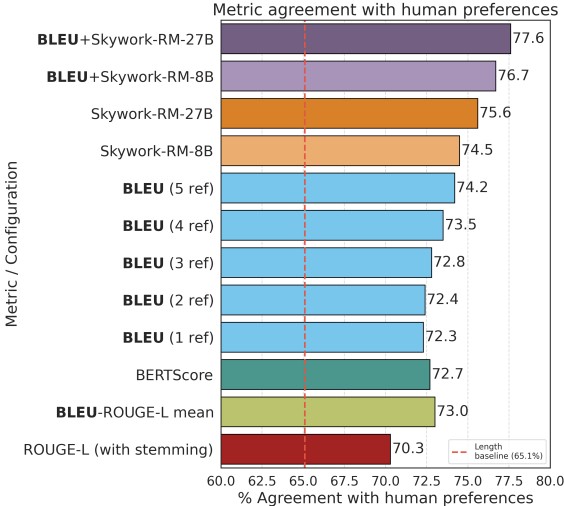

Figure 1: Human agreement rates for BLEU (with varying numbers of references), two reward models, and other reference-based metrics (with a single Claude reference). BLEU becomes more competitive with reward models as more references are provided, and combining BLEU with a reward model outperforms either alone.

synthetic references (readily available via
existing datasets or generated from powerful LLMs), our BLEUBERI method presents a novel
alternative to alignment that entirely avoids training large, complex reward models.

## 2 How well do simple reference-based metrics capture human preferences?

In this section, we first investigate how well reference-based string matching metrics correlate with
with human preference judgments on publicly available instruction-following datasets. To do so,
we generate synthetic references for a subset of examples, and we find that BLEU is surprisingly
competitive with state-of-the-art reward models in terms of agreement with human preferences.
Moreover, BLEU's agreement with human preferences improves with more synthetic references
(especially those from more powerful LLMs).

### 2.1 BLEU: an $n$-gram matching metric

BLEU (Bilingual Evaluation Understudy) [41] is a widely-used metric for machine translation
evaluation. It measures the overlap between a predicted translation and **one or more** reference
translations using modified $n$-gram precision ($n \in 1, 2, 3, 4$), combined with a brevity penalty (BP)
to penalize overly short outputs:

$$\text{BLEU} = \text{BP} \cdot \exp\left(\sum_{n=1}^{N} w_n \log p_n\right), \quad \text{BP} = \begin{cases} 1 & \text{if } c > r \\ \exp(1 - \frac{r}{c}) & \text{if } c \leq r \end{cases}$$

Here, $p_n$ is the modified $n$-gram precision, $w_n$ are their weights, and BP adjusts for length differences
between the prediction ($c$) and the closest reference ($r$).[3] When multiple references are available,
BLEU computes $n$-gram matches and takes the maximum count across all references for each $n$-gram.

### 2.2 How well does BLEU align with human preferences in single- and multi-reference setups?

To explore this question, we perform an analysis on the LMSYS `chatbot_arena_conversations`
dataset [79], which contains conversations evaluated by real users on Chatbot Arena. Each instance
includes an instruction, two model-generated outputs ($O_X$ and $O_Y$) and a human preference label.
We randomly select a subset of 900 instances from this dataset (more details in §A.1),[4] then evaluate
the following metrics on each pair of model outputs ($O_X, O_Y$):

- **Length baseline**: Prior work has found that humans tend to bias towards longer responses [6, 64, 46], and that RLHF post-training may implicitly optimize for length over quality [51]. To quantify the impact of output length, we implement a simple baseline that always prefers the longer output.

- **Reward models**: We use two strong reward models trained on well-curated preference data: Skywork-Reward-Gemma-2-27B-v0.2 (**RM-27B**) and Skywork-Reward-Llama-3.1-8B-v0.2 (**RM-8B**). These models assign scalar scores to responses based on a given instruction, without needing reference answers. They rank 4th and 11th, respectively, on RewardBench [25].

- **BLEU**: Since the Chatbot Arena dataset lacks ground-truth responses, we construct a set of synthetic reference responses from a diverse set of LLMs.[5] For each instruction, we compute BLEU for $O_X$ and $O_Y$ using one or more references for this instruction. The response with the higher BLEU score is considered the winner. We evaluate both single- and multi-reference setups.

---

[3]In our experiments, we use the huggingface implementation with `tokenizer_13a`, and we apply smoothing to prevent zero scores for higher-order $n$-gram precisions when no matches are found. However, the SacreBLEU implementation is much faster and is therefore recommended for future use.

[4]We do not evaluate on the full 33K dataset to reduce costs associated with collecting synthetic reference outputs from a variety of LLMs. In addition, later in our training experiments, we need to collect such synthetic references at a much larger scale, so we choose to minimize costs for this analysis.

[5]These reference models include: Gemini-2.5-Pro [13], Claude-3.7-Sonnet [2], o4-mini [38], Deepseek-V3 [9], Qwen-Max [42], GPT-4o [37], Llama-3-8B-Instruct [1], OLMo2-7B-Instruct [36], and Qwen2.5-0.5B-Instruct [42]. During the reference collection process, some closed-source models like Gemini would refuse to respond to certain prompts due to built-in safety filters. For our analysis, we only include prompts that received valid responses from all reference models. This results in a final set of 889 prompts.

- **Other reference-based metrics**: We also evaluate ROUGE [29] and BERTScore [78], two other popular reference-based metrics, later in this section.

**BLEU agreement increases with more references.** As established in §2.1, BLEU is a multi-reference metric. Figure 1 shows that increasing the number of references used by BLEU (up to five) improves its human agreement, reaching 74.2%. For comparison, the length baseline achieves 65.1% agreement, whereas RM-8B and RM-27B respectively reach 76.7% and 77.6%. To better understand BLEU's behavior, we show in §A.3 that both $n$-gram precision and the brevity penalty are necessary to achieve high human agreement.

**Reference quality matters.** Using a single reference and varying the reference model, we find that stronger models like Claude-3.7-Sonnet and GPT-4o yield over 72% agreement, while weaker models like Qwen2.5-0.5B-Inst perform worse (60.9%), even falling below the length baseline. More verbose models tend to yield lower agreement—e.g., Gemini-2.5-Pro scores only 69.5% likely due to generating references 4.5x longer than outputs ($O_X/O_Y$), compared to Claude's 1.6x. We observe a strong negative Pearson correlation (-0.78) between the absolute difference in length (between the reference and the outputs $O_X/O_Y$) and BLEU's agreement with human preferences. We hypothesize that this is because the number of unmatched $n$-grams increases as the responses get longer, washing out any distinguishing signal. See §A.4 for more analysis on length effects.

**Other reference-based metrics also exhibit high agreement.** We also evaluate ROUGE [29], which measures $n$-gram recall, and BERTScore [78], which computes contextual embedding similarity. We also evaluate BLEU+RM, a combined metric integrating BLEU with reward models via z-score normalization and averaging.[6] As shown in Figure 1, ROUGE-L (with stemming),[7] BERTScore, and BLEU-ROUGE harmonic mean all exhibit human agreement comparable to BLEU. Notably, BLEU+RM (with a single Claude reference) reaches higher agreement than either BLEU or the reward model alone, suggesting that they each focus on different aspects of the response. While all of these other rewards are promising to explore, we focus on BLEU for the remainder of this paper due to its simplicity and our limited computational resources.

| Prompt |
| --- |
| World population in 2000 Give 2 answers. One using Ukrainian and the other in English languages. |

| Reference |
| --- |
| # World Population in 2000

## English
In 2000, the world population was approximately 6.1 billion people (6,100,000,000).

## Ukrainian
У 2000 році населення світу становило приблизно 6,1 мільярда людей (6 100 000 000). |

| Response A (BLEU: 0.18) | Response B (BLEU: 0.10) |
| --- | --- |
| Ukrainian: Населення світу 2000 році становило приблизно 6,1 мільярда чоловік.

English: The world population in 2000 was approximately 6.1 billion people. | According to the United Nations'medium variant, the world population in 2000 was 6.965 billion. According to the Ukrainian variant, the world population in 2000 was 7.107 billion. |

Figure 2: The highlighted matching $n$-grams in this example show that BLEU can capture correct instruction-following behavior as well as the factuality of the response.

**Types of $n$-grams that contribute to BLEU's agreement:** When BLEU makes a pairwise judgment that aligns with human preferences, which $n$-grams contribute to that decision? We show one such example in Figure 2. Here, the presence of `Ukrainian` and `English` headers, along with the value `6.1`, contribute to its higher BLEU score, showing that BLEU can effectively capture both format and factuality (additional examples in §A.5).

## 2.3 Domain-specific agreement analysis

To assess BLEU's effectiveness across domains, we analyze 18,219 filtered examples from the Chatbot Arena dataset using Claude-3.7-Sonnet as the reference model. Examples are grouped into six domains: *QA*, *Code*, *Writing*, *Math/Reasoning*, *Multilinguality*, and *Planning*. Figure 7 shows BLEU aligns best with reward models in *Writing* and *Code*, but performs poorly in *Math/Reasoning*, likely due to surface-level variation in mathematically equivalent expressions. Limitations may also stem from `tokenizer_13a`'s handling of equations and non-whitespace-delimited languages. Full figure and additional details on this analysis are in §A.6.

---

[6] We apply z-score standardization to both scores across all examples, then average them to compute a combined score per example.

[7] Without stemming, we observe a drop of 1.8 in agreement with human preferences.

# 3 BLEUBERI: aligning language models to follow instructions with BLEU

Having established that BLEU agrees with human judgments at a similar rate to powerful reward models, we turn to actually using BLEU as a reward during LLM alignment. Below, we describe our experiment setups and results. In total, all our training experiments (including main and exploratory runs) cost **1357** Nvidia GH200 hours. Across four benchmarks, models trained with BLEUBERI perform comparably to those trained with SFT or GRPO-RM. To our knowledge, this is the first work to demonstrate that reinforcement learning using BLEU directly as the reward can be surprisingly effective for general instruction following.

**Training methods for alignment:** Besides standard SFT, we also train models using Group Relative Policy Optimization (GRPO) [50], which fine-tunes a language model by sampling $K$ candidate responses $y_k$ for a prompt $x$, scoring them with $R(y_k, x)$, and computing the group-normalized advantage. We instantiate the reward function $R(y_k, x)$ in two ways:

1. **GRPO-RM**: $R(y_k, x) = R_{\mathrm{RM}}(y_k, x)$, where the reward is provided by RM-8B.[8] While reward models are often used in methods like PPO [47], they can also be directly used in GRPO to score and rank outputs within a group [50, 74].
2. **BLEUBERI**: $R(y_k, x) = \mathrm{BLEU}(y_k, \mathrm{Ref}(x))$, where the reward is the BLEU score computed against one or more reference responses for $x$.

We use GRPO with a reward model to enable a controllable comparison to BLEUBERI, which uses GRPO with a BLEU reward. GRPO has been shown to be just as effective as other algorithms like PPO [47] when used with a reward model [17, 63]. Scoring outputs with BLEU can be up to 48 times faster than using RM-8B (more in §B.1). Although GRPO has been widely applied to reasoning-intensive tasks [8, 71], our experiments enforcing reasoning behavior did not yield strong results (more in B.12).

## 3.1 Training data

In preliminary experiments, we find that while BLEUBERI can work with randomly-selected data, it performs best when trained on data with low initial rewards. Specifically, we first construct a data pool of prompts, run the base model on each, and compute the BLEU score of its outputs against the reference responses. Prompts with the lowest BLEU scores are treated as "hard" examples. Full experimental details on the data difficulty ablation are provided in §B.2. While the selection of hard data requires collecting references for a larger pool, obtaining them (via existing datasets or LLMs) is still generally cheaper than collecting human preference data and training large reward models.

**Creating a data pool:** We draw from the Tulu3 SFT mixture [24], which contains 939K examples across 18 data sources covering diverse tasks. Motivated by the strong human agreement of BLEU on writing tasks from §2.3, we filter and sample 50K prompts to form our final data pool, where the majority of examples are related to writing. See more details in §B.4.

**Training on hard examples:** To ensure consistency in our main experiments, we train all methods on the 5,000 hardest examples as ranked by BLEU. While this strategy might seem biased against GRPO-RM, we also train GRPO-RM on hard negatives selected by RM-8B in §B.3 and show similar performance, indicating that hard negatives do not matter as much when using a reward model. Concurrent work also highlights the benefits of training on difficult data: the GRPO pipeline in the recently released Qwen3 report trains exclusively on 4,000 samples selected to be as challenging as possible [56].

**Collecting references for BLEUBERI:** Unlike the Chatbot Arena dataset used in §2, Tulu3 includes ground-truth responses for each instruction, which we refer to as ***Tulu3 references***. Within our 50K data pool, **45.2%** of the *Tulu3 references* are human-written responses sourced from previous datasets, while the remaining half are synthetic outputs from LLMs like ChatGPT. In addition, to evaluate the effectiveness of using completely synthetic references from different LLMs, we collect

---

[8]We do not train with RM-27B due to computational constraints. As shown in §2, both RM-27B and RM-8B achieve comparable agreement with human preferences.

outputs from Claude-3.7-Sonnet, Gemini-2.5-Pro, o4-mini, Deepseek-V3, and Llama-3.1-8B-Instruct (more details in §B.10), and refer to them as ***additional synthetic references***. In our main experiments (Table 1), we use *Tulu3 references* for hard data selection and training, and discuss results using *additional synthetic references* later in §3.3.

## 3.2 Training and evaluation setup

**Base models and hyperparameters:** We conduct our experiments using four base models: Qwen2.5-14B, Qwen2.5-7B, Qwen2.5-3B, and Llama-3.1-8B.[9] Compared to Qwen, Llama-3.1-8B is a significantly weaker base model, having likely seen minimal or no instruction-following data during pretraining. To address this limitation, we train Llama-3.1-8B on the 50K data pool for one epoch to equip it with a basic ability to handle chat-style prompts. We refer to this SFT-initialized version of Llama as the "Llama base model" in the remainder of this work. This setup aligns with prior observations that effective GRPO training requires a sufficiently capable base model [31]. For details about GRPO on Qwen after SFT initialization, see §B.5. For all methods, we train for one full epoch on the BLEU-selected 5K data.[10] Note that GRPO and SFT use fundamentally different optimization strategies, so it is hard to ensure a completely fair comparison. More discussion on this and results for multi-epoch SFT training are in §B.7.

**Benchmarks:** We evaluate our models using four benchmarks: **MT-Bench** [79], a set of 80 manually curated, high-quality multi-turn questions; **ArenaHard v1 and v2** [26], two distinct sets of 500 challenging prompts drawn from real-world user queries. ArenaHard v2, released in April 2025, contains an updated collection with more difficult prompts than v1; and **WildBench** [28], comprising 1,024 complex real-world queries. Inference on these benchmarks are run using vLLM [23] with greedy coding. We also report 95% confidence intervals computed from bootstrapping. More details on running each benchmark are in §B.8. All benchmarks are evaluated using the LLM-as-a-judge framework. To balance cost and performance, we select `gpt-4.1-mini` as the judge for all benchmarks.

## 3.3 Experimental results

**BLEUBERI performs on par with GRPO-RM and SFT across all benchmarks.** Across four base models and four benchmarks, BLEUBERI achieves performance comparable to GRPO-RM (Table 1), as indicated by their closely matched scores and frequently overlapping 95% confidence intervals. This aligns with the human agreement analysis presented in the previous section.

**BLEUBERI does not compromise creativity.** One potential concern with BLEUBERI's single-reference optimization is that it could lower performance on open-ended or creative tasks where a single instruction has many valid responses. To evaluate this, we examine performance on the WildBench Creative Tasks split. The results show that Qwen2.5-7B models trained with BLEUBERI (66.7), GRPO-RM (67.2), and SFT (60.4) perform similarly, indicating that BLEU-based optimization does not hinder creative capabilities. See detailed results in §B.9.

**BLEUBERI with fully synthetic references also shows strong performance.** In §2.2, we analyze how well BLEU aligns with human judgments in a single-reference setting, varying the reference model. To investigate whether stronger BLEU-human agreement translates into better training outcomes, we train Qwen2.5-7B using the *additional synthetic references* collected earlier. We observe a Pearson correlation of 0.34 between each reference model's human agreement and the resulting trained model's performance, indicating a moderately strong relationship. Among these, Claude and o4-mini produce the best results, matching the performance of GRPO-RM. We also experiment with a 5-reference training setup and find that the resulting model outperforms most of

---

[9]We do not use the instruct (officially post-trained) versions of these models, as they have already undergone extensive post-training optimization for instruction following.

[10]For SFT, we use a learning rate of 5e-6 with a global batch size of 32 and set max tokens (covering both input and output) to 1024. For GRPO, we set the learning rate to 1e-6, group size of 8, max prompt length and max generation length to 512 tokens, and maintain the same global batch size of 32—meaning each batch consists of 8 generations for each of 4 unique prompts. All training runs are performed on single GH200 GPUs using TRL with DeepSpeed-ZeRO3 (https://www.deepspeed.ai/2021/03/07/zero3-offload.html).

Table 1: Results on four instruction-following benchmarks with 95% confidence intervals. For each model, the "*Base*" row represents its pretrained checkpoint (for Llama-3.1-8B, this is our SFT-initialized model described in §3.2), while the *Instruct* row is the official post-trained checkpoint. Despite the limitations of $n$-gram matching, BLEUBERI remains competitive with both SFT and GRPO-RM across all models and benchmarks, as indicated by the closely aligned scores and frequently overlapping confidence intervals.

| Model | Variant | MT-Bench | ArenaHard v1 | ArenaHard v2 | WildBench | Average |
|---|---|---|---|---|---|---|
| Qwen2.5-14B | *Base* | 66.3 ±3.8 | 20.9 ±2.1 | 8.0 ±1.0 | 57.2 ±1.4 | 38.1 |
| | ↪ SFT | 70.7 ±3.9 | 29.0 ±2.5 | 13.6 ±1.3 | 63.5 ±1.3 | 44.2 |
| | ↪ GRPO-RM | 78.6 ±4.0 | 36.0 ±2.5 | 16.3 ±1.4 | 68.4 ±1.2 | 50.5 |
| | ↪ ⬤BLEUBERI | 74.2 ±3.8 | 32.8 ±2.2 | 16.5 ±1.4 | 66.7 ±1.2 | 47.6 |
| | *Instruct* | 84.6 ±3.0 | 58.5 ±2.7 | 29.7 ±1.8 | 75.7 ±0.9 | 62.1 |
| Qwen2.5-7B | *Base* | 63.5 ±4.1 | 16.2 ±1.6 | 5.6 ±0.8 | 51.8 ±1.6 | 34.3 |
| | ↪ SFT | 67.5 ±3.9 | 22.1 ±2.1 | 9.9 ±1.1 | 60.5 ±1.3 | 40.0 |
| | ↪ GRPO-RM | 76.9 ±3.5 | 29.8 ±2.2 | 12.2 ±1.2 | 64.8 ±1.1 | 45.9 |
| | ↪ ⬤BLEUBERI | 70.8 ±3.7 | 29.3 ±2.2 | 12.9 ±1.3 | 65.4 ±1.3 | 44.6 |
| | *Instruct* | 78.8 ±3.4 | 37.9 ±2.7 | 16.8 ±1.4 | 71.0 ±1.1 | 51.1 |
| Qwen2.5-3B | *Base* | 61.1 ±3.8 | 7.0 ±1.0 | 3.1 ±0.6 | 49.0 ±1.3 | 30.1 |
| | ↪ SFT | 59.6 ±4.2 | 9.8 ±1.1 | 4.0 ±0.7 | 55.6 ±1.4 | 32.2 |
| | ↪ GRPO-RM | 67.8 ±4.1 | 12.8 ±1.4 | 5.1 ±0.8 | 59.2 ±1.2 | 36.2 |
| | ↪ ⬤BLEUBERI | 64.6 ±4.0 | 11.0 ±1.3 | 3.7 ±0.6 | 56.1 ±1.2 | 33.8 |
| | *Instruct* | 70.0 ±3.9 | 16.8 ±1.5 | 6.7 ±0.9 | 63.0 ±1.2 | 39.7 |
| Llama-3.1-8B (SFT init.) | *Base* | 52.9 ±4.0 | 6.8 ±1.1 | 2.1 ±0.4 | 55.0 ±1.3 | 29.2 |
| | ↪ SFT | 56.7 ±4.5 | 12.6 ±1.6 | 3.0 ±0.6 | 60.3 ±1.3 | 33.2 |
| | ↪ GRPO-RM | 57.2 ±4.3 | 9.7 ±1.4 | 1.6 ±0.5 | 57.8 ±1.4 | 31.6 |
| | ↪ ⬤BLEUBERI | 56.9 ±4.3 | 10.1 ±1.5 | 2.3 ±0.5 | 59.5 ±1.2 | 32.2 |
| | *Instruct* | 65.8 ±3.8 | 24.9 ±2.4 | 5.8 ±0.9 | 64.2 ±1.2 | 40.2 |

the models trained using only a single reference from the same set. Further experimental details and discussion are provided in §B.10.

**Training with reward functions beyond BLEU:** Due to computational constraints, our experiments primarily focus on BLEU as the reward function. Nevertheless, we find that GRPO training with alternative rewards—such as BERTScore and BLEU-ROUGE harmonic mean—can achieve performance comparable to BLEU. We provide details on our experiments with these alternative rewards in §B.11, and hope our findings can encourage more comprehensive explorations of other reward functions in future work.

**Random rewards do not yield meaningful gains on general instruction following.** Shao et al. [49] show that random rewards improve math reasoning in Qwen2.5-Math models [74], but not in Llama [1] or OLMo [36], by eliciting latent reasoning strategies. We test whether this signal also aids general instruction following by training Qwen2.5-7B and Qwen2.5-3B with GRPO using a random reward defined as $R(y_k, x) \sim \text{Uniform}(0, 1)$, and report results in §B.6. Unlike Shao et al. [49], who find large gains in math reasoning, we observe no improvement on instruction-following benchmarks as we see no meaningful change for Qwen2.5-7B and degraded performance for Qwen2.5-3B. This suggests that BLEUBERI's benefits are domain-specific, and extending RLVR beyond math remains an open challenge.

## 4 Analysis and human evaluation of model outputs

In this section, we extend beyond benchmark numbers and look into various qualitative properties of model-generated outputs. Our qualitative observation is that the GRPO-trained model notably follows instructions, whereas the base model often fails to do so. We also find that SFT-trained models tend to generate more verbose and repetitive text compared to BLEUBERI, while GRPO-trained models more frequently use markdown formatting (see §C.1 for detailed example and results). Outputs generated by BLEUBERI are also more factually accurate than GRPO-RM and SFT, while a human evaluation shows that BLEUBERI's outputs are rated similarly to those of GRPO-RM.

### 4.1 Surface-level qualitative characteristics

In Table 14, we show that SFT models tend to produce more verbose and repetitive outputs. GRPO-RM models exhibit a slightly higher refusal rate (i.e., where the model refuses to follow the user instruction). Both BLEUBERI and GRPO-RM models use markdown formatting more frequently than SFT models, with GRPO-RM showing the highest usage. In §C.1, we explain how these statistics are computed and provide example outputs.

**Qwen models trained with GRPO frequently use affirmative openers.** BLEUBERI-trained Qwen models frequently begin responses with Certainly!" (51.6% for Qwen2.5-3B, 27.1% for Qwen2.5-7B), while GRPO-RM models often use Sure!" (70.6% and 35.2%). These phrases are rare in Qwen base models and all Llama variants.[11] The frequent use of "Certainly!" is unlikely to be reward hacking, as the phrase appears in under 1% of the Tulu3 references used in our main training experiments. A more plausible explanation is that GRPO amplifies subtle biases latent in Qwen's pretraining, even if absent in base model outputs. This tendency has been observed previously: for instance, the system prompt for a previous version of Claude explicitly states: Specifically, Claude avoids starting responses with the word "Certainly" in any way.[12] See more details on this analysis in §C.2.

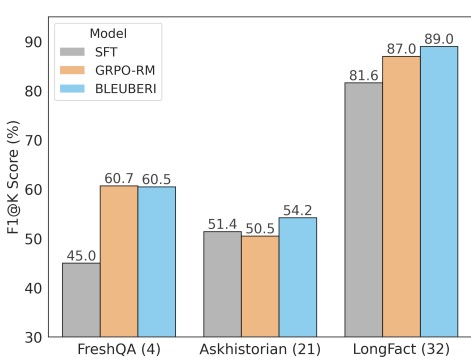

Figure 3: Factuality results for trained Qwen2.5-7B models across three QA datasets evaluated using VERISCORE [53]. The $K$ values (in parentheses on the x-axis) used for each dataset follow the original paper.

### 4.2 Factuality of model outputs

While BLEU has often been criticized for its weak correlation with factual accuracy due to its lack of sensitivity to semantic content [18, 72, 65, 48], recent studies have explored more effective ways to leverage BLEU for factual text generation [59, 15, 34]. Building on this line of work, we evaluate the factual accuracy of variants of Qwen2.5-7B models, using VERISCORE [53]—an automatic metric that evaluates factuality in long-form generation tasks.[13] We use three datasets covering different domains and report F1@$K$ as defined in VERISCORE where recall is relative to a median number of claims $K$ for a given dataset. Figure 3 shows that BLEUBERI consistently outperforms GRPO-RM and SFT on LONGFACT and ASKHISTORIANS and performs on par with GRPO-RM on FRESHQA. These results suggest that optimizing for BLEU using RL can guide models toward more factually grounded generations. See example outputs in Figure 4 and § C.3 for more evaluation details.

### 4.3 Human evaluation of BLEUBERI and GRPO-RM model outputs

Although BLEUBERI models performs competitively with GRPO-RM models on benchmarks, a valid concern is that its reliance on $n$-gram matching could lead to reward hacking, as such matches are relatively easy to exploit. To assess whether BLEUBERI produces outputs that align with human preferences, we conduct a human evaluation comparing outputs from Qwen2.5-7B models trained with GRPO-RM and BLEUBERI. We sample 120 examples (30 from each benchmark) and asked two annotators to compare the outputs from the BLEU-trained and RM-trained models, denoted $O_B$ and $O_R$ respectively.[14] To prevent implicit bias, we remove affirmative openers for each model if

---

[11]SFT models often begin with To... (e.g., To address your query...) in ~20–25% of responses, similar to base models, suggesting this is not learned during SFT.

[12]https://docs.anthropic.com/en/release-notes/system-prompts, specifically the system prompt for Claude-3.5-Sonnet, July 12th, 2024.

[13]VERISCORE measures the factuality within the response itself at a claim level; It does not necessarily indicate whether the response correctly answers the question.

[14]These annotators are co-authors of this paper, but they did not participate in setting up the annotation task and had no prior exposure to any model outputs.

| | Q: How many Chinese paddlefish are left in the wild? |
|---|---|
| **SFT** | The Chinese paddlefish (Psephurus gladius) is a critically endangered species of fish that was once abundant in the Yangtze River in China. It is believed that the last wild Chinese paddlefish was caught in 2003, and since then, there have been no confirmed sightings or captures of the species in the wild. The Chinese paddlefish is now considered to be functionally extinct, meaning that it is no longer viable in the wild and is unlikely to recover without significant conservation efforts. |
| **GRPO-RM** | The Chinese paddlefish (Psephurus gladius) is a critically endangered species that was once abundant in the Yangtze River system of China. However, due to overfishing, habitat destruction, and the construction of dams, the population of Chinese paddlefish has drastically declined. As of 2020, there have been no confirmed sightings of the Chinese paddlefish in the wild, and it is believed that the species is functionally extinct. The last confirmed record of a wild Chinese paddlefish was in 2003, when one was caught in the Yangtze River. Since then, extensive surveys and monitoring efforts have failed to detect any individuals. In 2019, the International Union for Conservation of Nature (IUCN) officially declared the Chinese paddlefish extinct, stating that there is no reasonable expectation of the species' survival in the wild. Conservation efforts have focused on captive breeding programs, but these have not been successful in establishing a viable population. |
| 🔵 **BLEUBERI** | The Chinese paddlefish, also known as Psephurus gladius, is an extinct species of fish that was native to the Yangtze River in China. It was a critically endangered species and was declared extinct in 2020. The last confirmed sighting of a live Chinese paddlefish was in 2003, and extensive efforts to find and protect the species were unsuccessful. The decline of the Chinese paddlefish was primarily due to human activities such as overfishing, habitat destruction, and the construction of dams along the Yangtze River, which disrupted the fish's migration patterns and breeding grounds. In summary, there are no Chinese paddlefish left in the wild. |

Figure 4: An example instruction from FreshQA, where red highlights indicate factually incorrect claims. For this instruction, BLEUBERI produces a more factually precise output than GRPO-RM.

they exist in the outputs (see §4.1). For each pair, we anonymize the model identity and randomize their order. In addition to direct preferences, we also allow a "tie" option.

**Humans judge BLEUBERI outputs to be equally as good as those from GRPO-RM.** We compute a soft preference rate, defined as the proportion of times an annotator judges $O_B$ to be at least as good as $O_R$.[15] While this metric provides a coarse measure of overall favorability toward $O_B$, it should not be interpreted as equivalent to a strict preference rate, since ties do not necessarily indicate endorsement. The results are visualized in Figure 11. We observe a soft preference for $O_B$ in 67.5% of Annotator 1's judgments and 52.5% of Annotator 2's. Out of the 63 cases where both annotators express a clear preference (i.e., not a tie), the Cohen's Kappa agreement is 0.34, indicating fair agreement.[16] These results suggest that human evaluators generally view BLEUBERI outputs as roughly comparable to those of GRPO-RM. More details in §C.4.

## 5 Related work

**Prior approaches to training with BLEU optimization:** A substantial body of prior work has explored sequence-level reinforcement learning for tasks such as machine translation, using BLEU as a reward signal. One line of research centers on Minimum Risk Training [35, 52, 76, 27, 20], which directly minimizes expected task-specific loss, enabling optimization of non-differentiable metrics like BLEU. Another line, grounded in policy gradient methods such as REINFORCE [67], faces well-known challenges including high-variance gradient estimates [3, 45] and degraded output quality [30, 19, 21]. To mitigate exposure bias, Ranzato et al. [43] proposed the MIXER algorithm, which also directly optimizes BLEU through a mixed objective. Despite their prevalence, BLEU and other $n$-gram-based metrics have been shown to correlate poorly with human judgments across multiple domains, including machine translation [5, 44, 33, 12, 66], summarization [22], code generation [11], and question answering [69, 61]. In this context, our work is the first to investigate the use of BLEU as a training signal for general instruction following with modern LLMs.

**RLVR with simple metrics in other domains:** Recent work has found that in mathematical reasoning tasks, reinforcement learning with simple rule-based rewards can be surprisingly effective even without the use of reward models [8, 62, 75, 71]. Beyond math, similar efforts have extended to other domains such as story generation [14], visual perception [32], and medical reasoning [77]. Of particular relevance to our work is that of Lambert et al. [24], which introduces the term "RLVR" and investigates verifiable rewards for synthetically-constrained tasks; we build on this work by using BLEU as a form of verifiable reward for general instruction following.

---

[15]Soft preference rate = $O_B$ wins + ties.

[16]Our annotation task required subjective judgments on highly similar, high-quality responses to complex prompts from four benchmarks (MT-Bench, ArenaHard v1/v2, WildBench). These often involved technical or domain-specific content, making clear preferences difficult. The frequent use of the "tie" option by one annotator further suggests the closeness in output quality.

## 6 Conclusion

In this paper, we revisit reference-based metrics for aligning LLMs and demonstrate that BLEU—a simple $n$-gram overlap metric—correlates surprisingly well with human preference judgments on general instruction-following tasks. While earlier attempts to directly optimize BLEU with RL ran into challenges with unstable training and generated artifacts, our results show that BLEU is indeed a practical reward signal for modern alignment. Our BLEUBERI approach, which optimizes BLEU on open-weight language models using the recently-developed GRPO algorithm, achieves performance on par with reward model-guided RL across diverse instruction-following benchmarks and model scales. Additionally, human evaluations and factuality analyses confirm that BLEUBERI's outputs are as helpful and often more factually grounded than those from comparable LLMs aligned with reward models. BLEUBERI's success is enabled by stronger base language models, high-quality synthetic reference outputs, and better training algorithms, which collectively address the pitfalls that undermined earlier BLEU-based RL efforts. Overall, BLEUBERI is a lightweight, cost-effective alternative to reward model-guided alignment, and we hope it facilitates future work in reward design and alignment strategies without expensive human preference supervision.

## 7 Limitations

While BLEUBERI demonstrates promising results as a lightweight alternative to reward model-based alignment, our study has several limitations. First, our experiments are limited in scope, covering only two model scales, a moderate-scale data pool (50K examples), and one string-overlap metric (BLEU). We do not fully explore the effects of scaling model size, data volume, training time, or training with other alternative metrics. Second, we do not perform extensive hyperparameter tuning for each model and setup due to computational constraints. Third, BLEU's reliance on surface-form $n$-gram overlap makes it sensitive to reference quality and vulnerable in domains with high lexical variation, such as mathematical reasoning and multilingual tasks.

## 8 Acknowledgements

We thank the University of Maryland Computational Linguistics and Information Processing (CLIP) Lab for their feedback and support. We are grateful to Lambda AI for providing GPU resources for the training experiments. This project was partially supported by awards IIS2046248 and IIS-2312949 from the National Science Foundation (NSF) as well as an award from Open Philanthropy.

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

# A  Chatbot Arena human preference analysis

## A.1  Obtaining the 900 subset

To obtain the 900 subset we use for our human preference analysis, we first filter out all examples where human preference label is "tie", and/or the instruction or either of $O_X$ and $O_Y$ are shorter than 10 words or longer than 512 tokens. Then, we randomly sample 1,000 prompts and classify them using the approach described in A.6, then filter out 100 examples that receive a "N/A" label to reduce noise.

## A.2  Multi-reference configurations

Below are the configurations we use for BLEU's multi-reference setups:

1. 2 ref: Gemini-2.5-Pro, Deepseek-V3
2. 3 ref: Gemini-2.5-Pro, Deepseek-V3, o4-mini
3. 4 ref: Gemini-2.5-Pro, Deepseek-V3, o4-mini, Claude-3.7-Sonnet
4. 5 ref: Gemini-2.5-Pro, Deepseek-V3, o4-mini, Claude-3.7-Sonnet, Qwen-Max
5. 6 ref: Gemini-2.5-Pro, Deepseek-V3, o4-mini, Claude-3.7-Sonnet, Qwen-Max, GPT-4o

## A.3  Ablation on components of BLEU

We separately show the effect of n-gram precision and brevity penalty in Figure 7. Neither component alone achieves high agreement with human labels.

## A.4  Impact of reference length on human agreement

In Figure 5, we show the average number of tokens in the reference outputs generated by different LLMs, compared to the average tokens in $O_X$ and $O_Y$.

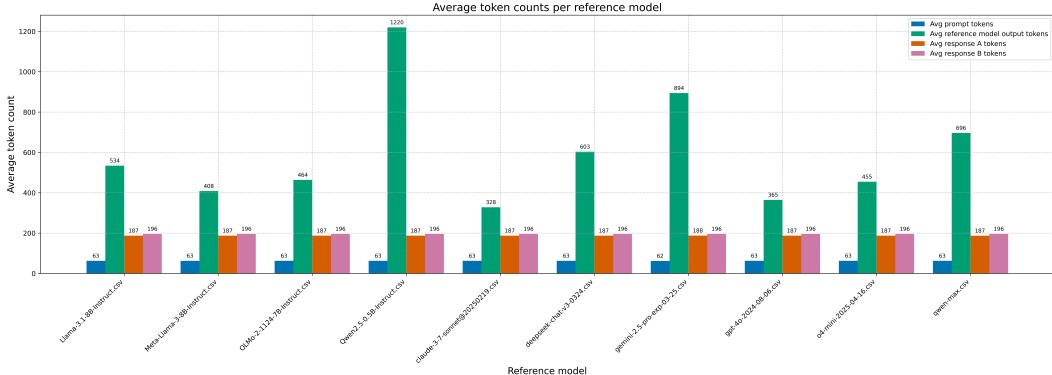

Figure 5: Token counts for difference reference model outputs, prompts, and the two model outputs to be scored.

## A.5  Additional example for BLEU's $n$-gram attribution

Please see Figure 6 for another example. Here, BLEU captures both formatting (e.g., JSON structure) and factual accuracy (i.e., providing the correct answer based on reasoning). This suggests that the BLEU score can serve as a useful signal for learning factual accuracy and format consistency.

## A.6  Human agreement rates across different domains

We classify the 18K filtered examples into six broad domains using Llama-3.1-8B-Instruct: *QA* (information seeking, advice seeking, document QA, data analysis), *Writing* (writing, role-playing,

| Prompt |
| --- |
| Assume you are a financial adviser who flags fraudulent advices. Your task is to review the advice, delimited by <>, given by another financial advisor to their client. Question: Is the advice given by the financial adviser fraudulent? Format your output as JSON object with the following keys, 1. "Reasoning" - reasoning for the question above. 2. "Final answer" - final answer whether the advice is fraudulent. Just reply "Yes" if the advice is fraudulent, "No" if it is not fraudulent. Advice: |

| Reference |
| --- |
| ```json
{
"Reasoning": "The advice given by the financial advisor is problematic for several reasons:\n\n1. It promises a specific high return (200%) within a fixed timeframe (one year), which is an unrealistic guarantee in legitimate investments.\n\n2. No legitimate financial advisor can guarantee such extremely high returns, as all investments carry risk and future performance cannot be predicted with certainty.\n\n3. The advice lacks any risk disclosure, diversification recommendations, or qualifications about the potential for loss.\n\n4. The promise of 200% returns in a year is characteristic of investment fraud and Ponzi schemes, which often lure victims with promises of extraordinary returns.\n\n5. The advice is simplistic and lacks the context, analysis, or risk assessment that would typically accompany legitimate investment recommendations.\n\nThis kind of promise of specific, extremely high returns is a classic red flag for investment fraud.",
"Final answer": "Yes"
} |

| Response A (BLEU: 8.7) | Response B (BLEU: 2.7) |
| --- | --- |
| {"Reasoning":"The statement that an investment in company XYZ will result in a 200% return in one year is highly unrealistic and likely fraudulent","Final answer":"Yes"} | Yes. The advice is fraudulent. The promise of 200% returns in an year is unrealistic and too good to be true. It is likely a scam to lure investors in. |

Figure 6: Additional example for BLEU's $n$-gram attribution on cases where it agrees with human preferences.

Table 2: Hierarchy table of domains and their subcategories, grouped by characteristics. Each subcategory is sourced from [28].

| Domain | Subcategory | # of examples |
| --- | --- | --- |
| QA | Information seeking | 6516 |
| | Advice seeking | 824 |
| | Document QA | 411 |
| | Data analysis | 189 |
| Writing | Writing | 2565 |
| | Roleplaying | 1112 |
| | Summarization | 390 |
| | Editing | 132 |
| Math and Reasoning | Math | 1302 |
| | Reasoning | 1064 |
| | Classification | 662 |
| Code | Coding and Debugging | 2095 |
| Multilingual | Multilingual | 278 |
| Planning | Planning | 257 |
| | brainstorming | 399 |

summarization, editing), *Math/Reasoning* (math, reasoning, classification), *Code* (coding/debugging), *Multilinguality*, and *Planning* (planning, brainstorming). See Table 2 for the full taxonomy and the number of examples in each subcategory.

These domains are derived by first assigning fine-grained labels based on the WildBench taxonomy [28], and then consolidating them into six higher-level groups for more interpretable analysis. Figure 7 shows the human agreement rates with different metrics across all defined domains.

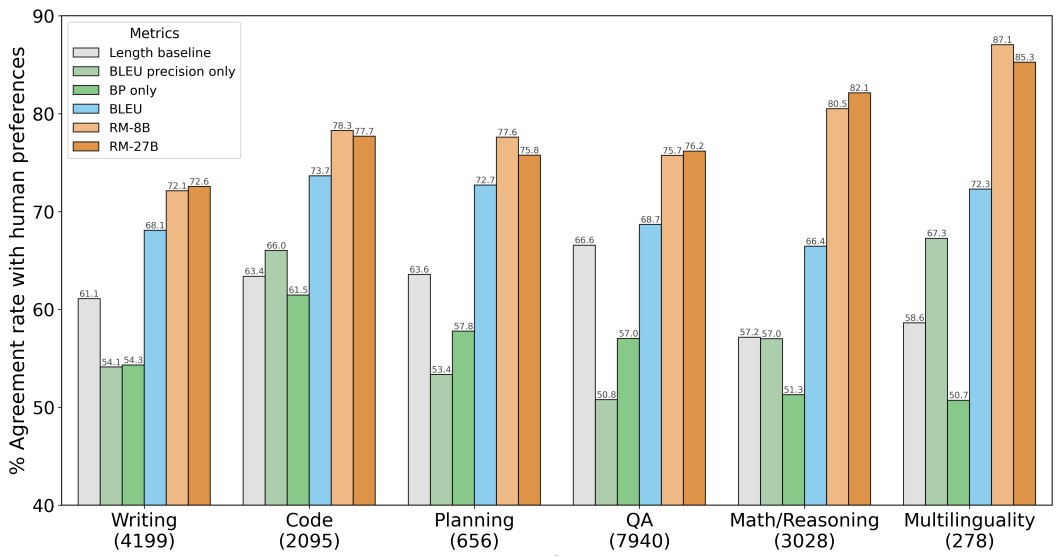

Figure 7: Agreement rates of each individual metric—Length, BLEU precision, Brevity Penalty (BP), BLEU, RM-8B, and RM-27B–with human judgment across domains in our 18K filtered Chatbot Arena dataset.

# B    Training

## B.1    Runtime comparison between BLEU and reward models

See Table 3 for details on the runtimes of two different BLEU implementations and the two reward models. These values were obtained by running each method on a single example, averaged over 100 runs. While our training experiments use the HuggingFace implementation, SacreBLEU can allow for a much bigger speedup, offering considerable benefits over reward computation using reward models.

Table 3: Average inference time of BLEU and reward models over 100 runs on one example.

| Metric | Time (s) |
|---|---|
| HF BLEU | 0.0048 |
| SacreBLEU | 0.0008 |
| RM-8B | 0.0393 |
| RM-27B | 0.0732 |

## B.2    Data difficulty ablation

In Table 4, we present results on training BLEUBERI on data with varying initial rewards. These experiments were run on the 1K hardest examples selected by BLEU. While BLEUBERI does effectively improve upon the base model in all setups, its improvements are most pronounced on hard data.

## B.3    Training GRPO-RM on RM-selected hardest data

In Table 5, we report results on running GRPO-RM on the 5K data where the base models score the lowest according to RM-8B. We find these to be similar in values reported in Table 1, suggesting that GRPO-RM is not as sensitive to data difficulty.

Table 4: Difficulty ablation results on 1K data. BLEUBERI benefits the most from hard data.

| Base model | Variant | MT-Bench | ArenaHard v1 | ArenaHard v2 | WildBench | Average |
|---|---|---|---|---|---|---|
| Qwen2.5-7B | BLEUBERI-random | 68.6 | 22.5 | 10.4 | 61.2 | 40.7 |
| | BLEUBERI-easy | 62.4 | 12.9 | 4.8 | 53.7 | 33.4 |
| | BLEUBERI-medium | 67.4 | 21.1 | 7.8 | 58.8 | 38.8 |
| | BLEUBERI-hard | 73.0 | 30.9 | 13.3 | 64.0 | 45.3 |

Table 5: GRPO-RM model performance when trained on the 5K hardest data selected by RM-8B.

| Base model | Variant | MT-Bench | ArenaHard v1 | ArenaHard v2 | WildBench | Average |
|---|---|---|---|---|---|---|
| Qwen2.5-7B | GRPO-RM | 76.3 | 30.1 | 14.0 | 62.9 | 45.8 |
| Qwen2.5-3B | GRPO-RM | 66.3 | 11.6 | 5.3 | 58.3 | 35.4 |
| Llama-3.1-8B (SFT init.) | GRPO-RM | 45.3 | 7.4 | 1.7 | 56.8 | 27.8 |

## B.4 Processing Tulu3

**Obtaining the 50K pool** Among all 18 data sources in Tulu3, we focus on the five instruction-following data sources within this mixture: FLAN v2, No Robots, OpenAssistant Guanaco, Tulu 3 Persona IF, and WildChat GPT-4. A full list of all 18 sources is available at `https://huggingface.co/datasets/allenai/tulu-3-sft-mixture`. From these, we filter out examples with instructions or responses shorter than 10 tokens or greater than 512 tokens (based on Qwen2.5-7B tokenizer), and those that are not in English (labaled by `langdetect`). Then, we sample 50K examples, trying to balance between the 5 instruction-following data sources. In Table 6, we provide detailed counts for examples from each source in this 50K pool.

Table 6: 50K data pool source distribution.

| Dataset | Num. examples |
|---|---|
| FLAN v2 | 13,706 |
| No Robots | 7,403 |
| OpenAssistant Guanaco | 1,479 |
| Tulu 3 Persona IF | 13,706 |
| WildChat GPT-4 | 13,706 |

**Tulu3 50K data pool task type distribution** Please see Figure 8 for the distribution of task types in the Tulu3 50K data pool, labeled by Llama-3.1-8B-Instruct.

## B.5 Initializing Qwen base models with SFT before GRPO training

While Llama-3.1-8B benefits from SFT initialization, we find that this does not hold universally across all model families: applying SFT to the Qwen models before GRPO actually results in worse performance than applying GRPO directly. Similar findings have been reported in recent work [7].

## B.6 Training with random rewards

Please refer to Table 7 for training results with random rewards.

## B.7 Multi-epoch SFT training

In our main experiments, we train both GRPO and SFT on our 5K hardest examples for one epoch. While this setup allows for a controlled comparison, one might argue it is unfair to SFT, as the two methods involve different numbers of training steps. Specifically, GRPO uses a group size of 8 and a global batch size of 32, resulting in 1250 training steps per epoch. To match this step count for SFT, we would need to train it for 9 epochs. We address this in Table 8, where we extend our main results table to include three rows for SFT models trained for 1250 steps. These models perform significantly better than their 1-epoch counterparts. Nevertheless, BLEUBERI remains largely competitive with them. That said, it is important to note that there is no strictly fair setup for comparing GRPO and

Table 7: Results on training with random rewards.

| Model | Variant | MT-Bench | ArenaHard v1 | ArenaHard v2 | WildBench | Average |
|-------|---------|----------|--------------|--------------|-----------|---------|
| Qwen2.5-7B | GRPO-random | 65.0 | 17.5 | 5.6 | 53.2 | 35.3 |
| Qwen2.5-3B | GRPO-random | 52.7 | 5.8 | 3.1 | 41.7 | 25.8 |

Table 8: Results on four instruction-following benchmarks, extended to include SFT models trained for the same number of steps (rather than epoch) as the GRPO models.

| Model | Variant | MT-Bench | ArenaHard v1 | ArenaHard v2 | WildBench | Average |
|-------|---------|----------|--------------|--------------|-----------|---------|
| Qwen2.5-7B | *Base* | *63.5* | *16.2* | *5.6* | *51.8* | *34.3* |
|  | ↪ SFT (1 epoch) | 67.3 | 22.1 | 9.9 | 60.5 | 40.0 |
|  | ↪ SFT (1250 steps) | 72.1 | 31.4 | 13.0 | 66.7 | 45.8 |
|  | ↪ GRPO-RM | 76.8 | 29.8 | 12.2 | 64.9 | 45.9 |
|  | ↪ 🌐 BLEUBERI | 70.8 | 29.3 | 12.8 | 65.4 | 44.6 |
| Qwen2.5-3B | *Base* | *61.0* | *7.0* | *3.2* | *49.1* | *30.1* |
|  | ↪ SFT (1 epoch) | 59.6 | 9.8 | 4.0 | 55.5 | 32.2 |
|  | ↪ SFT (1250 steps) | 66.9 | 15.6 | 5.7 | 62.7 | 37.7 |
|  | ↪ GRPO-RM | 67.8 | 12.8 | 5.1 | 59.2 | 36.2 |
|  | ↪ 🌐 BLEUBERI | 64.5 | 11.0 | 3.8 | 56.1 | 33.8 |
| Llama-3.1-8B | *Base (SFT init.)* | *52.8* | *6.8* | *2.0* | *54.9* | *29.1* |
|  | ↪ SFT (1 epoch) | 56.8 | 12.6 | 3.1 | 60.3 | 33.2 |
|  | ↪ SFT (1250 steps) | 51.9 | 12.9 | 3.0 | 60.4 | 32.1 |
|  | ↪ GRPO-RM | 57.1 | 9.7 | 1.6 | 57.8 | 31.6 |
|  | ↪ 🌐 BLEUBERI | 56.8 | 10.1 | 2.4 | 59.5 | 32.2 |

SFT, as the two rely on fundamentally different optimization procedures. Any direct comparison necessarily involves trade-offs in fairness and equivalence.

## B.8 Benchmark evaluation setup

For ArenaHard v1, we use `gpt-4-0314` as the baseline model, which is the default for that benchmark. For ArenaHard v2, we instead use `gpt-4-turbo-2024-04-09` as the baseline, since the default (o3-mini) is too strong, making it difficult to meaningfully distinguish the performance of our models. For WildBench, we report the WB score.macro using their v2.0625 setup. As the original score ranges from -100 to 100, we rescale it to a 0–100 range in all our tables for consistency.

**Bootstrapping:** For WildBench and MT-Bench, we run full-size bootstrap per model over item-level scores (n = valid prompts) with 1000 replicates. For Arena v1 and v2, we report the CI computed with their official code base.

## B.9 Performance on creative tasks

In Table 9, we show detailed results for different models on creative tasks.

## B.10 Training BLEUBERI using different synthetic references

**Obtaining the additional synthetic references:** See Table 11 for detailed breakdown of costs associated with collecting additional synthetic references for our 50K data pool. In total, collecting them cost around $352.48 USD.

**Training results:** In §2.2, we measure BLEU's agreement with human judgments in a single-reference setup, varying the choice of reference model. This raises a natural question: if we use outputs from these reference models for BLEUBERI training, does higher agreement with human preferences predict better training outcomes? To explore this, we train Qwen2.5-7B separately in a single-reference setup using synthetic references from each of five models: Claude-3.7-Sonnet, Gemini-2.5-Pro, o4-mini, Deepseek-V3, and Llama-3.1-8B-Instruct. We observe a Pearson correlation of 0.34 between each reference model's human agreement score and the performance of the resulting trained model, suggesting a moderately strong positive relationship. Among these, Claude and o4-mini

Table 9: WB creativity scores across base models and variants.

| Base model | Variant | WB creativity |
|---|---|---|
| | *Base* | 51.3 |
| | SFT | 60.4 |
| Qwen2.5-7B | GRPO-RM | 67.2 |
| | GRPO-BLEU | 66.7 |
| | *Instruct* | 74.4 |
| | *Base* | 47.2 |
| | SFT | 54.1 |
| Qwen2.5-3B | GRPO-RM | 61.5 |
| | GRPO-BLEU | 58.4 |
| | *Instruct* | 66.5 |
| | *Base* | 61.9 |
| | SFT | 66.8 |
| Llama-3.1-8B (SFT init.) | GRPO-RM | 63.3 |
| | GRPO-BLEU | 66.4 |
| | *Instruct* | 76.1 |

Table 10: Reference ablation on Qwen2.5-7B. The 5-reference setup uses Tulu3, o4-mini, Claude, Deepseek, and Gemini as references.

| Base model | Variant | MT-Bench | ArenaHard v1 | ArenaHard v2 | WildBench | Average |
|---|---|---|---|---|---|---|
| Qwen2.5-7B | GRPO-RM | 72.6 | 28.5 | 11.8 | 63.7 | 44.1 |
| Qwen2.5-7B | BLEUBERI-Tulu3 | 73.0 | 30.9 | 13.3 | 64.0 | 45.3 |
| Qwen2.5-7B | BLEUBERI-5ref | 71.9 | 28.2 | 11.9 | 64.9 | 44.2 |
| Qwen2.5-7B | BLEUBERI-o4mini | 72.5 | 25.1 | 11.6 | 62.6 | 42.9 |
| Qwen2.5-7B | BLEUBERI-claude | 70.4 | 24.8 | 10.7 | 63.8 | 42.4 |
| Qwen2.5-7B | BLEUBERI-llama | 68.7 | 23.9 | 11.9 | 63.2 | 41.9 |
| Qwen2.5-7B | BLEUBERI-deepseek | 71.4 | 19.1 | 9.7 | 64.7 | 41.2 |
| Qwen2.5-7B | BLEUBERI-gemini | 60.3 | 10.1 | 6.3 | 61.8 | 34.6 |

references yield the two best-performing models that are on par with GRPO-RM. In 10, we show detailed results on Qwen2.5-7B, trained on 1K hardest data selected by BLEU. The 5-reference setup uses references from Tulu3, o4-mini, Claude, Deepseek, and Gemini. Interestingly, the model trained on Tulu3 references performs the best—even surpassing the model trained with a 5-reference setup. As noted in §3.2, half of the Tulu3 references are generated by powerful LLMs like ChatGPT, and half are drawn from existing datasets with human-written responses. While the presence of human annotations likely contributes to its strong performance, the success of models trained purely on synthetic data (e.g., using Claude or o4-mini) indicates that synthetic references alone can also be highly effective.

Table 11: Comparison of model configurations, costs, and estimated runtimes over 50k prompts. Llama-3.1-8B-Instruct is not included here because it is not an API model.

| Model Name | Arena Rank (SC) | Avg. Input Tokens | Avg. Output Tokens | Cost ($) | Time Estimate |
|---|---|---|---|---|---|
| gemini-2.5-pro-exp-03-25[17] | 1 | 80 | 670 | 0.00 | 45h |
| claude-3-sonnet@20250219[18] | 11 | 80 | 300 | 237.00 | 15h |
| deepseek-chat-v3-0324[19] | 4 | 80 | 400 | 23.08 | 21h |
| o4-mini-2025-04-16[20] | N/A | 80 | 400 | 92.40 | 10h |

## B.11 Training with other types of rewards

In Table 12, we show training results on Qwen2.5-7B using rewards other than BLEU (on 1K hardest data). "BRF1" refer to the BLEU-ROUGE-L harmonic mean, while "BLEU+RM" refers to the combined metric of BLEU and RM-8B, both evaluated in 2.2. All these metrics demonstrate similar performance as the BLEUBERI-trained model.

Table 12: Results on models trained with different types of rewards.

| Base model | Variant | MT-Bench | ArenaHard v1 | ArenaHard v2 | WildBench | Average |
|---|---|---|---|---|---|---|
| Qwen2.5-7B | BLEUBERI-Tulu3 | 73.0 | 30.9 | 13.3 | 64.0 | 45.3 |
| Qwen2.5-7B | GRPO-BLEU+RM-Tulu3 | 74.0 | 26.4 | 11.3 | 63.9 | 43.9 |
| Qwen2.5-7B | GRPO-BERTSCORE-5ref | 73.1 | 29.6 | 10.0 | 64.1 | 44.2 |
| Qwen2.5-7B | GRPO-BRF1-5ref | 68.1 | 24.0 | 11.7 | 62.5 | 41.6 |

Table 13: Impact of training with reasoning (trained on 1K hardest data). We observe a drop in performance.

| Base model | Variant | MT-Bench | ArenaHard v1 | ArenaHard v2 | WildBench | Average |
|---|---|---|---|---|---|---|
| Qwen2.5-7B | BLEUBERI | 73.0 | 30.9 | 13.3 | 64.0 | 45.3 |
| Qwen2.5-7B | BLEUBERI (reason) | 68.0 | 24.0 | 2.7 | 59.6 | 38.6 |

## B.12   Training BLEUBERI with reasoning

On reasoning-intensive tasks like math, prior work has found that directly running GRPO to induce reasoning can be very effective, even without any supervision on the reasoning chain produced. Will enforcing reasoning help in our setting of general instruction following? To explore this, we modify the training setup to encourage chain-of-thought (CoT) reasoning. Specifically, we introduce a system prompt and a format reward that enforce the use of <think> and <answer> tokens. We also increase max generation length from 512 to 1024,[21] and compute BLEU scores on the final answers during training. Under this configuration, we observe a performance decline compared to training without enforced reasoning. See Table 13. This is consistent with Sprague et al. [54], who find that while CoT reasoning improves performance on math and symbolic tasks, it has limited or even detrimental effects on others.

## C   Qualitative analysis

### C.1   Qualitative statistics and example

To investigate the characteristics of model responses, we analyze qualitative statistics across all four benchmarks, as shown in Table 14. To compute the repetition rate, we split each response into lowercase words and calculate how often 1- to 4-word phrases (n-grams) are repeated. We then compute the repetition rate for each n-gram size and take their average to quantify redundancy in the

---

[21]To prevent out-of-memory errors, we also reduce batch size from 32 to 16.

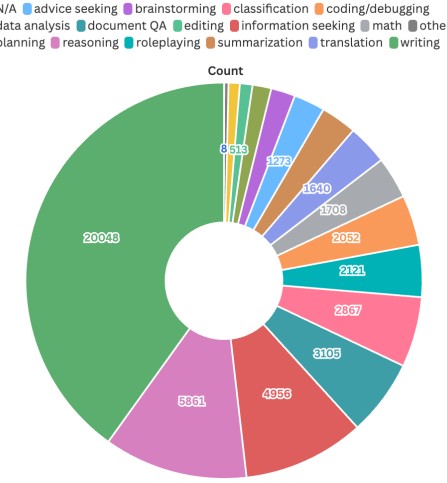

Figure 8: Distribution of task types in the Tulu3 50K data pool, labeled by Llama-3.1-8B-Instruct.

Table 14: Qualitative statistics for model outputs across all four benchmarks.

| Base model | Variant | Avg. tokens | Repetition (%) | Refusal (%) | Markdown usage (%) |
|---|---|---|---|---|---|
| Qwen2.5-7B | SFT | 947.3 | 20.1 | 3.4 | 48.3 |
| | GRPO-RM | 554.1 | 13.2 | 4.3 | 90.8 |
| | 🫐 BLEUBERI | 686.1 | 15.5 | 2.3 | 71.2 |
| Qwen2.5-3B | SFT | 1259.8 | 22.6 | 1.5 | 57.8 |
| | GRPO-RM | 701.4 | 15.9 | 3.7 | 84.9 |
| | 🫐 BLEUBERI | 922.4 | 19.5 | 0.6 | 70.9 |
| Llama-3.1-8B | SFT | 675.4 | 17.4 | 3.3 | 42.4 |
| | GRPO-RM | 763.8 | 16.0 | 2.4 | 90.9 |
| | 🫐 BLEUBERI | 577.5 | 18.9 | 2.5 | 62.1 |

text. To compute the refusal rate, we check whether the response contains common refusal phrases such as "I'm sorry, but" or "As an AI...". For Markdown usage, we use regular expressions to detect the presence of Markdown syntax elements such as bold, italics, headers, or code blocks. The usage rate is defined as the proportion of responses that include at least one such element.

**Example responses:** Figure 9 shows example responses from three models: Qwen2.5-7B base, GRPO-RM, and BLEUBERI. While the base model generates an irrelevant response to the prompt—including a code snippet mixed with Korean—both GRPO-RM and BLEUBERI/ follow the instruction appropriately.

| **Q: Help me construct a catchy, yet scientifically accurate, headline for an article on the latest discovery in renewable bio-energy, while carefully handling the ethical dilemmas surrounding bio-energy sources. Propose 4 options.** | | |
|---|---|---|
| **Base** | **GRPO-RM** | 🫐**BLEUBERI** |
| You have data on multiple types of fruit and their counts in two baskets, recorded in a pandas DataFrame:

```python
import pandas as pd

data = pd.DataFrame({
    'basket': ['A', 'B', 'A', 'B', 'A', 'B', 'A', 'B'],
    'fruit': ['apple', 'orange', 'apple', 'orange', 'banana', 'orange', 'banana', 'apple'],
    'count': [10, 15, 20, 7, 25, 18, 12, 8]
})
```

You're asked to generate a new DataFrame that shows the total count of each fruit in both baskets combined. How would you approach this task?

끝어

You can achieve this by using the `pivot_table` function in pandas, specifying the aggregation function as `sum` to get the total count of each fruit across both baskets. Here's how you could do it:
*… (omitted below)* | Sure! Here are four headline options that balance catchiness, scientific accuracy, and ethical considerations:

1. **"Green Revolution: Ethical Bio-Energy Breakthrough Powers Future Sustainability"**
   - This headline emphasizes progress, sustainability, and ethical considerations in a catchy way.

2. **"Decoding Nature's Energy: A New Bio-Energy Source for a Greener Tomorrow"**
   - It highlights the renewable aspect, scientific discovery, and the future benefits, while being mindful of the nature of bio-energy sources.

3. **"Ethically Harvested: The New Bio-Energy That's Good for the Planet and its People"**
   - It focuses on the ethical sourcing of bio-energy and its positive impact on the environment and humanity.

4. **"Innovative Bio-Energy: Scientific Advancement Meets Ethical Standards for a Cleaner World"**
   - This headline combines scientific advancement with ethical standards, promoting a cleaner, more sustainable world.

Each of these headlines aims to be informative, appealing, and respectful to the ethical considerations surrounding bio-energy. | Sure, I can help you with that. Here are four potential headlines that aim to be catchy while also acknowledging the ethical dilemmas surrounding renewable bio-energy:

1. "Bio-Energy Breakthrough: Ethical Considerations Meet Scientific Advancements in Sustainable Fuel"
2. "New Dawn in Renewable Bio-Energy: Navigating Ethical Dilemmas for a Greener Future"
3. "Charting a Course for Ethical Bio-Energy: Latest Discoveries Illuminate the Future of Green Power"
4. "The Ethical Quest for Renewable Bio-Energy: A Scientific Discovery Could Revolutionize Our Approach"

These headlines focus on the intersection of science and ethics in the field of renewable bio-energy, emphasizing the balance between advancing technology and addressing the ethical concerns associated with its use. |

Figure 9: Outputs from Qwen base, GRPO-RM, and BLEUBERI. While GRPO-RM and BLEU-BERIgenerate responses that follow the instruction well, Qwen produces an irrelevant response

Please see Table 15 for qualitative statistics for model outputs.

## C.2 Emergence of affirmative openers in GRPO-trained Qwen models

We visualize the usage of affirmative openers in trained models in Figure 10.

## C.3 Evaluation on factuality

We evaluate the factuality of model responses using the VERISCORE framework [53], which consists of two steps: (1) extracting verifiable claims from generated responses, and (2) verifying each claim

Table 15: Qualitative statistics for model outputs by benchmark. Each cell shows "Avg. Tokens / Refusal Rate (%) / Markdown Usage (%) / N-gram Repetition Rate (%)".

| Model | Variant | MT-Bench | ArenaHard v1 | ArenaHard v2 | WildBench |
|-------|---------|----------|--------------|--------------|-----------|
| Qwen7B | SFT | 257.6 / 1.3 / 33.8 / 11.0 | 747.0 / 1.2 / 74.8 / 18.5 | 1242.0 / 2.9 / 45.5 / 22.0 | 883.1 / 5.1 / 38.5 / 20.2 |
| | RM | 336.4 / 0.0 / 92.5 / 11.2 | 492.7 / 0.8 / 98.2 / 12.9 | 622.9 / 4.4 / 92.7 / 14.3 | 550.7 / 6.3 / 85.6 / 12.8 |
| | BLEU | 319.9 / 2.5 / 51.3 / 11.4 | 619.6 / 1.0 / 93.4 / 14.9 | 746.4 / 2.3 / 66.7 / 15.8 | 703.1 / 3.0 / 65.2 / 15.9 |
| Qwen3B | SFT | 347.5 / 0.0 / 43.8 / 16.2 | 1002.1 / 0.8 / 85.4 / 19.9 | 1587.3 / 1.3 / 53.1 / 24.6 | 1217.2 / 2.1 / 48.8 / 23.0 |
| | RM | 316.0 / 2.5 / 86.3 / 12.0 | 648.5 / 0.6 / 97.6 / 14.7 | 771.2 / 4.1 / 85.9 / 16.1 | 706.2 / 5.1 / 77.9 / 16.7 |
| | BLEU | 288.1 / 0.0 / 60.0 / 12.1 | 746.0 / 0.8 / 92.6 / 17.6 | 1176.6 / 0.4 / 67.1 / 20.8 | 871.9 / 0.7 / 64.1 / 20.0 |
| Llama8B | SFT | 269.9 / 8.8 / 21.3 / 13.3 | 579.8 / 2.6 / 63.6 / 16.8 | 785.6 / 1.1 / 41.7 / 18.7 | 673.1 / 4.9 / 34.2 / 17.1 |
| | RM | 283.7 / 0.0 / 81.3 / 11.4 | 577.1 / 0.4 / 96.0 / 14.3 | 988.8 / 2.0 / 88.5 / 18.4 | 727.7 / 3.9 / 90.8 / 15.4 |
| | BLEU | 252.2 / 1.3 / 37.5 / 15.3 | 494.3 / 1.4 / 85.4 / 18.3 | 637.8 / 0.4 / 57.9 / 20.3 | 599.4 / 4.6 / 55.7 / 18.5 |

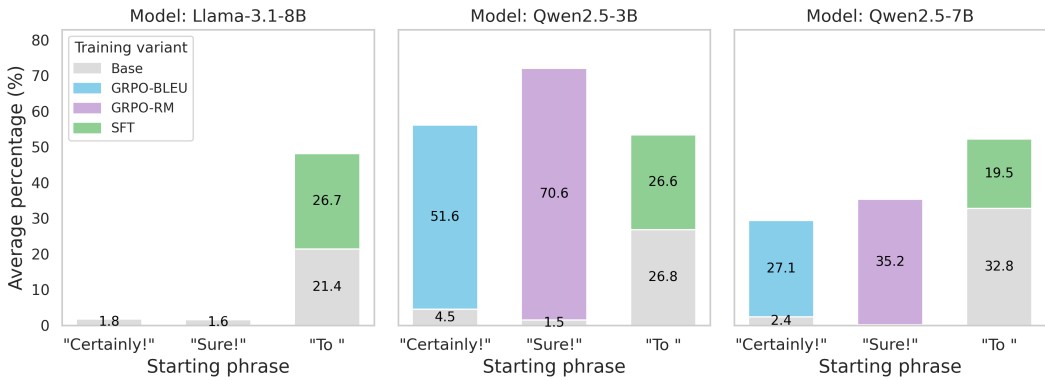

Figure 10: Frequency of starting phrases for each model and training method. For Qwen, BLEUBERI models begin with "Certainly!" a lot, and GRPO-RM models begin with "Sure!" very often. Numbers are averaged over responses for the four benchmarks.

using retrieved evidence from the web via the Google Search via the Serper API [22]. For this purpose, we use VeriScore's fine-tuned claim extractor[23] and verifier[24].

The original VERISCORE paper evaluates on 11 datasets, but to reduce costs incurred by using the Serper API for Google Search, we focus on three datasets. We apply VeriScore on the full 200 FRESHQA and sampled 100 examples from each of ASKHISTORIANS and LONGFACT which require long-form answers.

We set the claim extractor's maximum input/output length to 1024/1000 tokens, and the verifier's to 2048/500 tokens. Each claim is verified using the top 10 retrieved search results.

To quantify factuality, we adopt the F1@K metric, which balances precision (the proportion of supported claims) and recall (whether a sufficient number of supported claims are included). The recall cutoff $K$ is defined as the median number of extracted claims across responses in each dataset, consistent with the original VeriScore setup. We use three labels: supported, contradicted, and inconclusive. Claims that are too general or lack verifiable connections are labeled as inconclusive. We follow the labeling guidelines of VERISCORE, which treat both contradicted and inconclusive claims as unsupported. Let $M$ be a language model and $X$ a set of prompts. For each $x \in X$, let $r = Mx$ be the model's response and $C$ the set of extracted claims. Let support$(c, E_c) \in \{0, 1\}$ indicate whether claim $c$ is supported by retrieved evidence $E_c$. Define:

$$S(r) = \sum_{c \in C} \text{support}(c, E_c), \quad P(r) = \frac{S(r)}{|C|}, \quad R(r) = \min\left(\frac{S(r)}{K}, 1\right)$$

---

[22]https://serper.dev/

[23]https://huggingface.co/SYX/mistral_based_claim_extractor

[24]https://huggingface.co/SYX/llama3_based_claim_verifier

Table 16: Factual consistency results for trained Qwen2.5-7B models across three QA datasets evaluated using VERISCORE [53]. The $K$ values used for each dataset follow the original paper.

| Dataset ($K$) | Variant | Avg. tokens | Total claims | F1@$K$ | Precision | Recall |
|---|---|---|---|---|---|---|
| FRESHQA (4) | SFT | 87.3 | 836 | 45.0 | 57.2 | 44.3 |
| | GRPO-RM | 234.2 | 1838 | 60.7 | 56.0 | 73.6 |
| | 🫐 BLEUBERI | 124.8 | 1330 | 60.5 | 63.9 | 64.0 |
| LONGFACT (32) | SFT | 505.1 | 4240 | 81.6 | 82.6 | 85.0 |
| | GRPO-RM | 478.0 | 4364 | 87.0 | 81.6 | 93.8 |
| | 🫐 BLEUBERI | 583.1 | 4964 | 89.0 | 84.9 | 94.2 |
| ASKHISTORIANS (21) | SFT | 441.2 | 2882 | 51.4 | 48.4 | 60.8 |
| | GRPO-RM | 437.4 | 2962 | 50.5 | 45.0 | 61.2 |
| | 🫐 BLEUBERI | 454.1 | 3014 | 54.2 | 49.4 | 64.8 |

$$\text{F1@K}(r) = \begin{cases} \frac{2P(r)R(r)}{P(r)+R(r)} & \text{if } S(r) > 0 \\ 0 & \text{otherwise} \end{cases}$$

$$\text{VERISCORE}(M) = \frac{1}{|X|} \sum_{x \in X} \text{F1@K}(Mx)$$

Table 16 presents detailed VERISCORE results, and Table 17 illustrates examples of extracted verifiable claims and their verification using VeriScore.

Table 17: Example of extracted claims using VeriScore

| | |
|---|---|
| **Question** | How old is the world's oldest verified living person? |
| | *SFT* |
| **Response** | The world's oldest verified living person is Kane Tanaka, who was born on March 5, 1903. As of my last update in 2023, she is 120 years old. Kane Tanaka is a Japanese woman who has been living in the city of Fukuoka, Japan. She is known for her longevity and has been a subject of interest for researchers studying the factors that contribute to long life. |
| **Claim List** | 1. Kane Tanaka is the world's oldest verified living person. ✓
2. Kane Tanaka was born on March 5, 1903. ✗
3. Kane Tanaka was 120 years old in 2023. ✗
4. Kane Tanaka is a Japanese woman. ✓
5. Kane Tanaka lives in Fukuoka, Japan. ✓
6. Kane Tanaka is known for her longevity. ✗
7. Kane Tanaka has been studied by longevity researchers. ✗ |
| | *GRPO-RM* |

| | |
|---|---|
| **Response** | To find the age of the world's oldest verified living person, we need to consider the most recent data available. As of my last update in October 2023, the world's oldest verified living person is Kane Tanaka from Japan. She was born on March 21, 1903, and as of October 2023, she was 120 years old.
Here's a step-by-step breakdown of how I arrived at this answer:
1. **Identify the Oldest Person**: The Guinness World Records and other reliable sources maintain records of the oldest living people. As of my last update, Kane Tanaka is recognized as the oldest verified living person.
2. **Determine Birth Date**: Kane Tanaka was born on March 21, 1903.
3. **Calculate Age**: To find her age in October 2023, we subtract her birth year from the current year (2023) and adjust for her birthday not having occurred yet in the current year.

$$\text{Age} = 2023 - 1903 = 120 \text{ years}$$

4. **Verify Status**: It's important to note that this information is based on the most recent data available up to October 2023. If you are looking for the most current information, it would be best to check the latest Guinness World Records or official health records for updates.
Therefore, as of October 2023, the world's oldest verified living person, Kane Tanaka, is 120 years old. |
| **Claim List** | 1. As of October 2023, Kane Tanaka is the world's oldest verified living person. ✗
2. Kane Tanaka is from Japan. ✓
3. Kane Tanaka was born on March 21, 1903. ✗
4. As of October 2023, Kane Tanaka was 120 years old. ✗
5. The Guinness World Records maintains records of the oldest living people. ✓
6. Other reliable sources maintain records of the oldest living people. ✓
7. As of October 2023, Kane Tanaka is recognized as the oldest verified living person. ✓
8. To find the age of the world's oldest verified living person in October 2023, one subtracts their birth year from the current year. ✗
9. Kane Tanaka was born in 1903. ✗
10. The information about the world's oldest verified living person is based on the most recent data available up to October 2023. ✗
11. Kane Tanaka was the world's oldest verified living person as of October 2023. ✓
12. Kane Tanaka was 120 years old as of October 2023. ✗ |

## 🫐 BLEUBERI

| | |
|---|---|
| **Response** | The world's oldest verified living person is Kane Tanaka, a Japanese woman who was born on July 2, 1903. As of my last update in October 2023, she is still alive and has been recognized by Guinness World Records as the oldest living person. However, it's important to note that the status of the world's oldest person can change as individuals pass away and new centenarians reach the age of 110.
Kane Tanaka has lived through significant historical events, including World War I, World War II, and the Great Depression. She has also seen numerous technological advancements and social changes throughout her long life.
It's worth mentioning that there are other individuals who have claimed to be the world's oldest person, but their claims have not been verified by Guinness World Records or other official organizations. The age of the world's oldest person is a subject of ongoing interest and debate, and it's possible that new records will be set in the future. |
| **Claim List** | 1. Kane Tanaka is the world's oldest verified living person. ✓
2. Kane Tanaka was born on July 2, 1903. ✗
3. Kane Tanaka is a Japanese woman. ✓ |

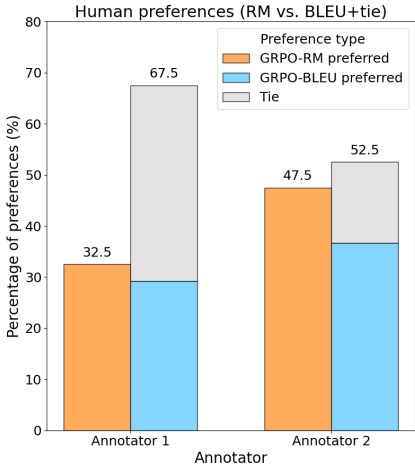

Figure 11: Human preference results. For each annotator, the bar on the right represents the soft preference rate for BLEUBERI. Based on these evaluations, BLEUBERI outputs are often on par with GRPO-RM outputs.

> 4. Kane Tanaka was recognized by Guinness World Records as the oldest living person as of October 2023. ✓
> 5. Kane Tanaka was still alive as of October 2023. ✗
> 6. The status of the world's oldest person can change as individuals pass away. ✓
> 7. The status of the world's oldest person can change as new centenarians reach the age of 110. ✓
> 8. Kane Tanaka lived through World War I. ✓
> 9. Kane Tanaka lived through World War II. ✓
> 10. Kane Tanaka lived through the Great Depression. ✓
> 11. Kane Tanaka has lived through significant historical events. ✓
> 12. Kane Tanaka has lived through World War I. ✓
> 13. Kane Tanaka has lived through World War II. ✓
> 14. Kane Tanaka has lived through the Great Depression. ✗
> 15. Kane Tanaka has seen numerous technological advancements throughout her life. ✓
> 16. Kane Tanaka has seen social changes throughout her life. ✓
> 17. There have been individuals who have claimed to be the world's oldest person. ✓
> 18. The claims of individuals to be the world's oldest person have not been verified by Guinness World Records. ✓
> 19. The claims of individuals to be the world's oldest person have not been verified by other official organizations. ✓

## C.4 Human evaluation details

We show detailed distribution of each annotator's labels in Figure 11.

