# OpenReview forum: "BLEUBERI: BLEU is a surprisingly effective reward for instruction following"
_NeurIPS.cc/2025/Conference — NeurIPS 2025 poster_

### Official Review · Reviewer_G3EU · 2025-06-18

**Clarity:** 3
**Significance:** 2
**Originality:** 3
**Rating:** 3
**Confidence:** 3

**Summary:**

This paper describes experiments using BLEU as the reward function for training models with GRPO. Experiments have been conducted on 4 instruction-following benchmarks and they find that effectiveness of using BLEU is contingent on being able to create high-quality references during training. Specifically, the authors use data from the Tulu v3 SFT mixture, and use 4 powerful models to create reference answers. Along with the use of human answers from the SFT dataset 4 LLMs are trained using BLEU as the reward function. Experiments conducted on MT Bench, Arena Hard v1 and v2, Wildbench suggest using BLEU as a reward function outperforms SFT and is marginally below using an existing LLama based reward model. Human evaluation studies on a small set of $120$ responses reveal that model outputs are comparable to RM-trained models.

**Questions:**

Q1. A surface-level metric like BLEU likely just steers the model towards responding "like" strong models which are already human-aligned. Is this really about BLEU -- isn't the use of n-gram overlap just saying, the model should prefer responses to be similar to the way the "frontier" models respond (if one uses the LLM references) or asking the model to "prefer the response that's  {\em more} `similar' to the reference answer? I note that the results in Table 1 do not use the additional synthetic references

Q2. Related to my first question, a baseline that feels missing is one that's based on an LLM as a judge -- tasked with the same/similar task as BLEU. Lets say we ask an LLM something like "which of these responses is closer/equivalent to the ground-truth reference"? Would we expect this to do even better than BLEU considering it might do a more meaningful comparison with a reference answer? And if it would, aren't we just saying GRPO can use a reward function that compares the relative goodness/closeness/alignment to a high quality reference as the training criterion?

**Ethical Concerns:**

["NO or VERY MINOR ethics concerns only"]

**Final Justification:**

The paper will benefit from a round of revision especially including baselines discussed during the rebuttal period. I would like to retain my score.

**Limitations:**

Discussing aspects related to the questions in the review may be a useful discussion.

**Quality:**

3

**Strengths And Weaknesses:**

- Strengths
  - Interesting idea to use a metric such as BLEU
  - Extensive experiments (including some interesting ones in the appendix)
- Weakness
   - See questions

---

> ### Author Rebuttal · Authors · 2025-07-31
>
> We appreciate the reviewer’s recognition of our work, including the interestingness of using BLEU as a reward signal for alignment, and our extensive experimental scope. Below, we respond to each of the weaknesses and questions in detail.
>
> ## Weaknesses (same as questions)
>
> ***`1. BLEU’s effectiveness is dependent on reference quality`***
> We acknowledge this limitation, but we also want to highlight that **obtaining high-quality references is relatively straightforward**. Such data can come from two sources:
>
> * **Synthetic outputs from frontier LLMs**: Generating synthetic data has become a standard component of modern LLM training pipelines (Lambert et al., 2024; Qwen Team, 2024). These synthetic outputs have proven to be highly effective in practice.
> * **Ground-truth responses from existing datasets**: Numerous publicly available datasets already provide high-quality reference answers. Prominent examples include Tulu 3 (Lambert et al., 2024), OpenOrca (Mukherjee et al., 2023), UltraChat (Ding et al., 2023), and Alpaca (Wang et al., 2022).
>
> In contrast, **training reward models demands extensive human-labeled preference data, which is costly and time-consuming**. Reference-based approaches like BLEUBERI offer a more accessible and efficient path to alignment, demonstrating that even simple metrics like BLEU can serve as practical alternatives to reward models.
>
> References:
>
> * Tulu 3: Pushing Frontiers in Open Language Model Post-Training (Lambert et al., 2024\)
> * Qwen2.5 Technical Report (Qwen Team, 2024\)
> * Enhancing Chat Language Models by Scaling High-quality Instructional Conversations (Ding et al., 2023\)
> * Orca: Progressive Learning from Complex Explanation Traces of GPT-4 (Mukherjee et al., 2023\)
> * Self-Instruct: Aligning LM with Self Generated Instructions (Wang et al., 2022\)
>
> ***`2. A baseline that feels missing is one that’s based on an LLM as a judge -- tasked with the same/similar task as BLEU`***
> We agree that using an LLM-as-a-judge to compare responses against a reference could, in principle, provide a more semantically nuanced reward than BLEU. However, incorporating it during training introduces potential drawbacks that may undermine the primary appeal of reference-based metrics (simplicity and efficiency):
>
> * LLMs are much more expensive to run than BLEU and similar metrics like ROUGE. Additionally, if the LLM judge generates long-form justifications, it will be much slower than reward models.
> * Reference-based metrics like BLEU provide a fully transparent and deterministic reward signal, whereas LLM-based judgments are inherently opaque.
>
> To clarify, our focus on BLEU throughout the paper stems from its unique position among reference-based metrics: **it has a particularly simple form, yet it can already be surprisingly effective**. Our intent is not to argue that BLEU is the optimal reference-based reward function. Rather, we demonstrate that **even such a basic metric can rival the performance of learned reward models (which are essentially finetuned LLM judges)**. We hope this insight can encourage further exploration of other reference-based metrics.

---

> > ### Comment · Reviewer_G3EU · 2025-08-04
> > **Thank you**
> >
> > Thank you for responding to my review. I think my main concerns hold -- primarily whether the reference based metric is just a roundabout way of just asking the model to "prefer" responses from a powerful LLM teacher. Secondly, to my point about the LLM Judge -- modern LLMs are fairly good at gauging semantic equivalence. It wont be too complicated to setup up a yes-no judge (even with some output processing)

---

> > > ### Author Response · Authors · 2025-08-04
> > >
> > > We thank the reviewer for considering our rebuttal and for providing further feedback!
> > >
> > > While recent work has started to use LLM judges to compute the rewards during RL training (Gunjal et al., 2025), our work shows that simpler and faster metrics like BLEU can accomplish surprisingly strong results. We believe that this result is novel compared to prior work. Using lightweight metrics is much less expensive, and could enable better scaling.
> > >
> > > In the final version of the manuscript, we will include experiments using an LLM judge and compare its performance and efficiency to BLEU-based training.
> > >
> > > References:
> > > - Rubrics as Rewards: Reinforcement Learning Beyond Verifiable Domains (Gunjal et al., 2025)

---

> > > > ### Comment · Reviewer_G3EU · 2025-08-05
> > > > **Follow up**
> > > >
> > > > Agree that a cheap reward/score is always a good thing but its unclear whether the model is ending up learning something other than "respond in text that is `similar' to reference text" and this could have also been posed as a (simpler) preference problem. And then there's a question of whether BLEU/ROUGE or equivalent is really the ideal scorer? An LLM judge is definitely more expensive than BLEU but is it much better? Considering the model is trained with on-policy RL how much more expensive would a light LLM call to check for semantic equivalence have been?
> > > >
> > > > The paper would benefit from experiments and discussions about these; it feels incomplete without such critical analysis.

---

> > > > > ### Author Response · Authors · 2025-08-05
> > > > >
> > > > > We sincerely appreciate the followup questions and address them point by point below:
> > > > >
> > > > > `1. It's unclear whether the model is ending up learning something other than "respond in text that is 'similar' to reference text"`
> > > > >
> > > > > We’d like to clarify that training a model to respond in a style similar to high-quality references is a non-trivial challenge, **especially given the broad diversity of tasks covered by general instruction-following datasets**. While BLEU does reward surface-level overlap, our analysis (Sec. 2, Fig. 2) shows that the n-grams it favours frequently encode factual content and correct formatting, not just lexical similarity. Our evaluations also indicate that this type of training does lead to improvements across benchmark scores, human preference ratings, and factual accuracy.
> > > > >
> > > > > `2. Is BLEU/ROUGE or equivalent really the ideal scorer?`
> > > > >
> > > > > Our claim is **not** that BLEU is the best possible reward signal. Instead, our central goal is to show that **a lightweight, transparent metric like BLEU can guide GRPO training to match the performance of much heavier learned reward models**. Learned reward models are essentially LLM judges fine-tuned on large amounts of labeled preference data; they encode opaque, implicit preferences and operate without explicit references. In contrast, BLEU offers a more transparent alternative whose reward depends only on an inspectable reference set.
> > > > >
> > > > > `3. Considering the model is trained with on-policy RL how much more expensive would a light LLM call to check for semantic equivalence have been?`
> > > > >
> > > > > We understand this concern and will add a detailed comparison between BLEU-based training and a lightweight LLM-judge reward in the revised manuscript, including wall-clock latency and cost.
> > > > >
> > > > > We again thank the reviewer for the thoughtful feedback and look forward to incorporating these clarifications!

---

> > > > > > ### Comment · Reviewer_G3EU · 2025-08-06
> > > > > > **Acknowledgement**
> > > > > >
> > > > > > Thank you again for engaging in the discussion. I wanted to acknowledge that I have read your response and noted the points made.

---

> > > > > > > ### Author Response · Authors · 2025-08-06
> > > > > > > **Thank you for the engagement**
> > > > > > >
> > > > > > > We really appreciate the reviewer engaging in multiple rounds of discussion. To briefly summarize our core contribution: we show that lightweight reference-based metrics like BLEU can achieve performance comparable to that of heavily trained reward models when used for GRPO training. We believe this novel finding lays important groundwork for future exploration of lightweight metrics in alignment research.
> > > > > > >
> > > > > > > In response to the reviewer's valuable feedback, we will include experiments using an LLM-based judge in the next version of the paper, as suggested. Additionally, we will quantify the differences in speed and quality compared to our BLEU-based models. We would appreciate if the reviewer could kindly consider raising their score based on the discussion.

---

### Official Review · Reviewer_wK8y · 2025-06-29

**Clarity:** 3
**Significance:** 3
**Originality:** 3
**Rating:** 4
**Confidence:** 4

**Summary:**

This work presents a new post-training algorithm that uses the BLEU score on references as the reward. The BLEUBERI reward can be used in any RL optimization algorithms, including GRPO. The proposed method is evaluated on multiple tasks with two models (Qwen 2.5 and Llama 3.1). Models trained with BLEUBERI show similar or better performance compared to SFT or GRPO with normal RM rewards. This work also contains interesting analyses showing BLEUBERI leads to more grounded responses, and the method is also compatible with synthetic data or other reference-based metrics.

**Questions:**

1. BLEUBERI with synthetic data feels like some sort of distillation on a high level. Have you compared it with distillation methods empirically?
2. What will happen if you do not filter out easy data when training with BLEUBERI? Are there some common reward hacking patterns?
3. In the agreement study in Sec. 2, the best method is to use BLEU + RM. Have you tried similar approaches during training? Will it also lead to the best performance?
4. How much markdown formatting do you have in your training data? Does that explain the trends in Sec. 4.1?
5. I don't get the "amplifies subtle biases" part in line 291. Can you elaborate?

**Ethical Concerns:**

["NO or VERY MINOR ethics concerns only"]

**Final Justification:**

The main idea of this paper is enough interesting so I'd inclined to accept this paper despite that there are real room for improvements for this paper (as pointed out by me and other reviewers). The authors provide a good response to my review, but it does not fully address my comment in weakness 1 (Qwen results are still much better than Llama results), so I'll keep my score unchanged.

**Limitations:**

yes

**Quality:**

3

**Strengths And Weaknesses:**

## Strengths
1. Using BLEU as the reward during post-training is an original but natural idea. It's interesting to see that its performance is good and even surpasses SFT or RLHF with RM in some cases.
2. This paper is well-written, and the analyses in this work are thorough. I like how this work starts with an agreement study and also complements the experiments with a number of interesting analyses (e.g., experiments with synthetic data, other reference-based metrics, and qualitative analysis on behavior difference).

## Weaknesses
1. This work is tested on two different models: Qwen 2.5 and Llama. The performance on Qwen is really good. However, the trend on Llama is quite different. The different trends across different models are a bit concerning, as one may wonder if the advantages of using BLEUBERI are an artifact of the Qwen model. This work currently contains a few experiments, only focusing on Qwen models. Will the findings be the same if they are conducted on Llama models?
2. This is not really a weakness of the current work, but I hope this work can provide more insights into how different post-training methods lead to different behaviors. Using reference-based metrics as a reward feels like a middle ground between SFT (which enforces copying every token in the reference) and RL with RM (which does not tie closely to the reference at all). I'm super curious about how these supervisions with different "strengths" lead to different behavior. While the current work already provides some interesting case studies, there are no conclusive or systematic findings yet.

---

> ### Author Rebuttal · Authors · 2025-07-31
>
> We appreciate the reviewer’s recognition of our work, including the originality of using BLEU as a reward signal for alignment, the clarity of our writing, and the thoroughness of our analyses, including the agreement study, synthetic data experiments, alternative metrics, and qualitative behavior analysis. Below, we respond to each of the weaknesses and questions in detail.
>
> ## Weaknesses
>
> ***`1. One may wonder if the advantages of using BLEUBERI are an artifact of the Qwen model.`***
> We acknowledge that Qwen is a much stronger base model than Llama, and thus it is easier to get RL to work on the Qwen models compared to Llama. Prior work has observed similar findings in RL experiments in other domains (Shao et al., 2025). In our experiments, we apply a brief stage of SFT to Llama before running RL to provide the model with a basic understanding of instruction-style data. A more detailed discussion can be found in **Section 3.2, Line 218-223**. Importantly, our results in **Table 1** show that on Llama-3.1-8B (with SFT initialization), BLEUBERI *can* achieve competitive performance with GRPO-RM and SFT, suggesting that **the effectiveness of BLEUBERI is not merely an artifact of the Qwen base model**.
>
> References:
>
> * Spurious Rewards: Rethinking Training Signals in RLVR (Shao et al., 2025\)
>
> ***`2. More insights into how different post-training methods lead to different behaviors.`***
> We agree that using reference-based metrics as a reward is like a middle ground between SFT and RL with reward models. To summarize, our current findings suggest that:
>
> * SFT produces the most verbose and repetitive outputs, followed by BLEUBERI, then GRPO-RM (**Table 13**).
> * BLEUBERI-trained models produce more factually grounded outputs than SFT-trained and GRPO-RM-trained models (**Section 4.2**).
>
> To the best of our knowledge, our work is the first to explore applying simple reference-based metrics to improve general-domain instruction following. We hope these findings **lay the groundwork for more systematic explorations** into how different forms of supervision shape model behavior in instruction following tasks.
>
> ## Questions
>
> ***`1. BLEUBERI with synthetic data feels like some sort of distillation on a high level. Have you compared it with distillation methods empirically?`***
> We agree that BLEUBERI shares a high-level resemblance to knowledge distillation. In our current experiments, SFT acts as the closest analog to distillation, though it does not incorporate teacher logits in the way classical distillation methods do. As shown in Table 1, BLEUBERI consistently outperforms SFT across benchmarks. This is likely because **SFT tends to induce broad behavioral shifts, whereas GRPO facilitates more targeted, fine-grained adjustments** (Rajani et al., 2025).
>
> In the next revision of the manuscript, we plan to include a standard distillation baseline for comparison.
>
> References:
>
> * Scalpel vs. Hammer: GRPO Amplifies Existing Capabilities, SFT Replaces Them (Rajani et al., 2025\)
>
> ***`2. What will happen if you do not filter out easy data when training with BLEUBERI? Are there some common reward hacking patterns?`***
>
> We report results from training with BLEU on easy, medium, and randomly sampled data in Table 4 (Appendix B.2). While random sampling improves over the base model, training on the hardest examples yields significantly better performance. This is in line with findings from prior work (Qwen Team, 2025).
>
> Regarding common reward hacking patterns, we do observe that models trained on easy data tend to produce more repetitive outputs compared to those trained on harder examples. We will try to quantify this better in the next version.
>
> References:
>
> * Qwen3 Technical Report (Qwen Team, 2025\)
>
> ***`3. In the agreement study in Sec. 2, the best method is to use BLEU + RM. Have you tried similar approaches during training? Will it also lead to the best performance?`***
> Yes, we report training results using the combined BLEU+RM reward in **Table 11 (Appendix B.10)**. Interestingly, while this reward showed the highest agreement with human preferences in the evaluation study (Section 2), the BLEUBERI model slightly outperforms the BLEU+RM-trained model in terms of benchmark performance.
>
> For the BLEUBERI and BLEU+RM models in Table 11 (both trained on 1,000 examples), we performed bootstrapping to estimate confidence intervals (please refer to our response to Reviewer R3fD for details). The performance difference between the two models appears modest, as their confidence intervals frequently overlap. Furthermore, we did not tune the reward weights in the BLEU+RM setup, which could leave room for further optimization.
>
> | Base model   | Variant         | MT-Bench | 95% CI         | ArenaHard v1 | 95% CI         | ArenaHard v2 | 95% CI         | WildBench | 95% CI         | Average |
> |--------------|----------------|----------|----------------|--------------|----------------|--------------|----------------|-----------|----------------|---------|
> | Qwen2.5-7B   | BLEUBERI        | 73.1     | (-3.7, \+3.7)   | 30.9         | (-1.7, 2.3)     | 13.3         | (-1.2, \+1.2)   | 64.0      | (-1.2, \+1.3)   | 45.3    |
> | Qwen2.5-7B   | GRPO-BLEU+RM    | 74.1     | (-3.4, \+3.4)   | 26.4         | (-1.8, 2.7)     | 11.4         | (-1.2 / \+1.4)  | 63.9      | (-1.3, \+1.2)   | 43.9    |
>
> ***`4. How much markdown formatting do you have in your training data? Does that explain the trends in Sec. 4.1?`***
> Among the 5,000 training examples used for each model in Table 1, here is the percentage of data where the reference response contains markdown formatting:
>
> * Qwen2.5-7B: 21.8%
> * Qwen2.5-3B: 24.0%
> * Llama-3.1-8B (SFT init.): 13.8%
>
> While markdown is not overwhelmingly dominant in the training data, its presence likely contributes to the trends in Section 4.1 (Table 13). GRPO-based methods may amplify features that correlate with higher rewards (e.g., structured formatting), so even modest amounts of markdown in the training data could be reinforced. In contrast, SFT matches references token-by-token, which may underemphasize sparse formatting cues, leading to lower markdown usage in outputs.
>
> ***`5. I don’t get the “amplifies subtle biases” part in line 291. Can you elaborate?`***
> In line 291, “amplifies subtle biases” refers to the phenomenon observed in Zhao et al. (2025), where RL post-training tends to magnify preferences or stylistic patterns already weakly present in the pretrained model, even if they are not strongly expressed in the base model outputs. We will revise this sentence in the next version and add the appropriate citation for clarity.
>
> References:
>
> * Echo Chamber: RL Post-training Amplifies Behaviors Learned in Pretraining (Zhao et al., 2025\)

---

> > ### Comment · Reviewer_wK8y · 2025-08-06
> >
> > Thank you for your response which resolves most of my questions. However, while I understand that the effectiveness of BLEUBERI is not merely an artifact of the Qwen base model, I do still have concerns on how well most of the Qwen results and analyses in the paper will generalize to different models. Therefore, I will maintain my original score of 4.

---

### Official Review · Reviewer_dhgB · 2025-07-02

**Clarity:** 4
**Significance:** 3
**Originality:** 3
**Rating:** 4
**Confidence:** 4

**Summary:**

This paper presents BLEUBERI, which uses BLEU as a reward signal during GRPO instead of expensive reward models for alignment for instruction following. Despite that BLEU is just a simple n-gram matching metric, this paper demonstrates that it can be surprisingly effective in matching the performance of sophisticated reward models for instruction-following tasks.

**Questions:**

1. What would happen if experimented with ROUGE (more recall focused) instead of BLEU (more precision focused), or use both?

2. how sensitive is it to use BLEU with different hyper-parameters, e.g., different n-gram length?

**Ethical Concerns:**

["NO or VERY MINOR ethics concerns only"]

**Limitations:**

yes

**Paper Formatting Concerns:**

no issues

**Quality:**

3

**Strengths And Weaknesses:**

Strengths:
- BLEU is efficient to compute compared to calling expensive reward models, thus the empirical findings of this paper is valuable to the broader empirical community.
- Despite its simplicity, BLEUBERI-trained models perform comparably to more expensive approaches relying on reword models across four challenging benchmarks (MT-Bench, ArenaHard v1/v2, WildBench) and three different base models (Llama-3.1-8B, Qwen2.5-7B, Qwen2.5-3B).
- The authors also demonstrate that the closer guidance based on BLEU signals leads to models that are more factually grounded than those trained with traditional reward models or supervised fine-tuning, as measured by VeriScore factuality evaluation.

Weaknesses:
- This approach is applicable only when high-quality references are available to compute BLEU against.
- The strengths still outweigh the above weakness, given that collecting reference data can be often more cost-effective than calling the reward models, and one can also leverage already existing datasets with references.

---

> ### Author Rebuttal · Authors · 2025-07-31
>
> We appreciate the reviewer’s recognition of our work, including BLEU’s computational efficiency compared to reward models, the strong performance of BLEUBERI-trained models across four benchmarks and multiple base models, and the improved factual grounding achieved through BLEUBERI training. Below, we respond to each of the weaknesses and questions in detail.
>
> ## Weaknesses
>
> ***`1. This approach is applicable only when high-quality references are available to compute BLEU against.`***
> ***`2. The strengths still outweigh the above weakness, given that collecting reference data can be often more cost-effective than calling the reward models, and one can also leverage already existing datasets with references.`***
> We acknowledge this limitation, but as the reviewer also noted, **obtaining high-quality references is relatively straightforward**. Such data can come from two sources:
>
> * **Synthetic outputs from frontier LLMs**: Generating synthetic data has become a standard component of modern LLM training pipelines (Lambert et al., 2024; Qwen Team, 2024). These synthetic outputs have proven to be highly effective in practice.
> * **Ground-truth responses from existing datasets**: Numerous publicly available datasets already provide high-quality reference answers. Prominent examples include Tulu 3 (Lambert et al., 2024), OpenOrca (Mukherjee et al., 2023), UltraChat (Ding et al., 2023), and Alpaca (Wang et al., 2022).
>
> In contrast, **training reward models demands extensive human-labeled preference data, which is costly and time-consuming**. Reference-based approaches like BLEUBERI offer a more accessible and efficient path to alignment, demonstrating that even simple metrics like BLEU can serve as practical alternatives to reward models.
>
> References:
>
> * Tulu 3: Pushing Frontiers in Open Language Model Post-Training (Lambert et al., 2024\)
> * Qwen2.5 Technical Report (Qwen Team, 2024\)
> * Enhancing Chat Language Models by Scaling High-quality Instructional Conversations (Ding et al., 2023\)
> * Orca: Progressive Learning from Complex Explanation Traces of GPT-4 (Mukherjee et al., 2023\)
> * Self-Instruct: Aligning LM with Self Generated Instructions (Wang et al., 2022\)
>
> ## Questions
>
> ***`1. What would happen if experimented with ROUGE (more recall focused) instead of BLEU (more precision focused), or use both?`***
> We currently discuss alternative metrics like ROUGE-L in Section 2.2 and Appendix B.10. Below, we provide some **new experiment results** with ROUGE-L and a BLEU \+ ROUGE-L ensemble metric on the Chatbot Arena analysis:
>
> | \# references | ROUGE-L (no stemming) | ROUGE-L (with stemming) | BLEU & ROUGE-L mean |
> |--------------|------------------------|--------------------------|----------------------|
> | 1            | 68.5%                  | 70.3%                    | 73.0%                |
> | 2            | 72.3%                  | 73.9%                    | 74.4%                |
> | 3            | 72.7%                  | 74.1%                    | 75.5%                |
> | 4            | 71.7%                  | 72.2%                    | 75.5%                |
> | 5            | 72.2%                  | 72.9%                    | 75.0%                |
>
> From these new results, we find that **with stemming enabled** and **more references available**, ROUGE-L can achieve human agreement comparable with BLEU. Combining BLEU and ROUGE-L also yields strong agreement results. We will add and discuss these results in the next version.
>
> ***`2. How sensitive is it to use BLEU with different hyper-parameters, e.g., different n-gram length?`***
> We present **new experiment results** below, which shows that BLEU performance is relatively stable across different maximum n-gram orders, as long as the order is greater than 1\.
>
> | max ngram order | alignment |
> |-----------------|-----------|
> | 1               | 69.3%     |
> | 2               | 71.5%     |
> | 3               | 72.9%     |
> | 4               | 72.4%     |
> | 5               | 71.2%     |
> | 6               | 71.5%     |
>
> For the standard setup with max ngram order set to 4, we also experimented with both smoothed and unsmoothed BLEU variants and found that **using smoothing is crucial** for achieving high human agreement. With smoothing, the agreement is 72.4%. Without smoothing, the agreement drops to 54.7%. Smoothing prevents BLEU scores from collapsing to zero when higher-order n-gram matches are sparse, even in outputs that are meaningfully aligned. In all experiments in the paper, we apply smoothing (as noted in Footnote 2 in the paper). We will add and discuss these results in the next version.

---

> > ### Author Response · Authors · 2025-08-07
> > **Followup**
> >
> > Dear Reviewer dhgB,
> >
> > Thank you once again for your thoughtful review. If you have any follow-up questions or concerns regarding our rebuttal, we would be happy to provide additional clarification before the discussion period concludes.
> >
> > To summarize, we have included **new analysis results with ROUGE-L and ROUGE-L + BLEU ensemble**, **analysis on BLEU's sensitivity to hyperparameters**, and provided clarifications for other points you raised.
> >
> > If our rebuttal has addressed your concerns, we would be grateful if you would consider adjusting your score accordingly. We sincerely appreciate your time and helpful feedback!

---

### Official Review · Reviewer_dqb5 · 2025-07-04

**Clarity:** 4
**Significance:** 2
**Originality:** 2
**Rating:** 4
**Confidence:** 4

**Summary:**

This paper investigates the feasibility of using the BLEU metric as a reward for the GRPO algorithm in instruction-following alignment. As a pivotal study on metric effectiveness, the authors show that BLEU’s agreement with human judgment improves as the number of references increases or when it is combined with other metrics. In the main experiments, BLEUBERI performs comparably to GRPO-RM across most settings. Notably, BLEUBERI demonstrates potential advantages in improving factuality while maintaining creativity.

**Questions:**

I will be insterested to see the results on larger base model sizes (32B) with the same reward models comparison (RM-8B vs BLEUberi). If the results are positive for BLEUberi, this would be a strong demonstration for light weight reward function.

A more direct quantification for GRPO algorithm speed-up between RM-8B and BLEU might be presented.

**Ethical Concerns:**

["NO or VERY MINOR ethics concerns only"]

**Final Justification:**

The authors have partially addressed my concerns, which has led me to revise my rating from 'Borderline reject' to 'Borderline accept'

**Limitations:**

yes

**Quality:**

3

**Strengths And Weaknesses:**

Strengths

1. The paper revealed that Reward Models are costly for alignment GRPO and discovered that BLEU is surprisingly effective reward for instruction following. This topic has rarely discussed before and provides interesting findings to the community.

2. The paper adopted synthethic data from tulu3, which might potentially usefule to researchers. Also, the paper has done an comparison of metric's effectivity, which produces insight to the community.

Weaknesses


1. Limited Analysis of Reward Hacking Risks. While the factual presentation is appreciated, the discussion of potential reward hacking remains limited. Upon reviewing the Appendix, I observed that BLEU used as a reward exposes models to two notable risks:

Verbosity Bias: As shown in Table 13, BLEUBERI causes a marked increase in generated output length, indicating a verbosity issue.

Repetition Bias: Table 13 also illustrates that BLEUBERI leads to higher repetition in generated text, a common failure mode for text generation models.

These factors raise significant concerns about the effectiveness of BLEU as a reward metric for instruction following. A more nuanced analysis—including mechanisms to detect and mitigate these biases—would improve the discussion.

2. Insufficient Discussion of Alternative Rule-Based/Verifiable Metrics. More thorough analysis on rule-based/verifiable metrics. as presented in section2, Appendix B.10, there should be chances to in-depth discussion. It is hard for me to recognize why ROUGE-L, BERTScore, BLEUrt are not competitive to BLEU. Without such, it is hard to assess BLEU’s relative contribution as a low-cost reward. Also, the emsemble of rewards should also be considered.

3. Concerns with Human Evaluation Reporting (line 320): The soft preference rate is misleading. One cannot calculate the tie rate into the 'for' rate. The same problem exists in Appendix Figure 11. Besides, Cohen Kappa of 0.36 for triple selection indicates the observed two annotators agreement is around 45%, which does not demonstrate strong reliability for the human evaluation results. Emphasizing these limitations and contextualizing the implications for the overall conclusions would be prudent.

---

> ### Author Rebuttal · Authors · 2025-07-31
>
> We appreciate the reviewer’s recognition of our work, including BLEUBERI as a cost-effective alternative to GRPO-RM for alignment, the novelty and interestingness of our findings, the value of our collected synthetic data, and the informative comparison of different metrics. Below, we respond to each of the weaknesses and questions in detail.
>
> ## Weaknesses
> | **Base Model** | **Variant** | **Avg. Tokens** | **Repetition (%)** |
> | -------------- | ----------- | --------------- | ------------------ |
> | Qwen2.5-7B     | SFT         | 947.3           | 20.1               |
> |                | GRPO-RM     | 554.1           | 13.2               |
> |                | BLEUBERI    | 686.1           | 15.5               |
> | Qwen2.5-3B     | SFT         | 1259.8          | 22.6               |
> |                | GRPO-RM     | 701.4           | 15.9               |
> |                | BLEUBERI    | 922.4           | 19.5               |
> | LLaMA-3.1-8B   | SFT         | 675.4           | 17.4               |
> |                | GRPO-RM     | 763.8           | 16.0               |
> |                | BLEUBERI    | 577.5           | 18.9               |
>
> These numbers suggest that **BLEUBERI is not consistently more verbose than GRPO-RM, and overall differences between the two are relatively modest**.
>
> On length bias:
>
> * We note that our benchmarks try to mitigate length bias by design. Both ArenaHard v1 and v2 apply style control, which control for factors like length and markdown usage. WildBench similarly has a length penalty mechanism, which converts small wins/losses into ties if one response is much longer than the other.
>
> On repetition bias:
>
> * As shown in Table 13, BLEUBERI has lower repetition rates than SFT and rates that are generally close to—or only slightly higher than—those of GRPO-RM. Despite this, our human evaluation (Section 4.3) finds **BLEUBERI model outputs to be just as preferable as GRPO-RM’s**. This indicates that the minor verbosity and repetition seen in Table 13 do not substantially affect the perceived quality of responses.
>
> ***`2. Insufficient Discussion of Alternative Rule-Based/Verifiable Metrics`***
> We currently discuss alternative metrics in Section 2.2 and Appendix B.10. We agree that further exploration of such metrics could provide valuable insights. **Our primary goal, however, was to demonstrate that even simple metrics like BLEU can be effective.** Given our resource constraints, we focused on BLEU in the training experiments due to its simplicity and surprisingly strong performance. To the best of our knowledge, **our work is the first to apply simple reference-based metrics to enhance general-domain instruction following**. We hope this serves as a foundation for future research on rule-based or verifiable reward functions.
>
> *`2.1. “It is hard for me to recognize why ROUGE-L, BERTScore, BLEUrt are not competitive to BLEU. Also, the ensemble of rewards should also be considered.”`*
> ROUGE-L and BLEU \+ ROUGE-L ensemble **(new experiments)**
>
> * We ran additional experiments on the Chatbot Arena dataset with ROUGE-L. With stemming enabled and multiple reference responses, **ROUGE-L can achieve human agreement levels comparable to BLEU**. Moreover, **combining BLEU and ROUGE-L (with stemming) as an ensemble reward function by averaging also yields strong agreement results**. These findings will be included and discussed in the next version of the manuscript.
>
> | \# references | ROUGE-L (no stemming) | ROUGE-L (with stemming) | BLEU & ROUGE-L mean |
> |--------------|------------------------|--------------------------|----------------------|
> | 1            | 68.5%                  | 70.3%                    | 73.0%                |
> | 2            | 72.3%                  | 73.9%                    | 74.4%                |
> | 3            | 72.7%                  | 74.1%                    | 75.5%                |
> | 4            | 71.7%                  | 72.2%                    | 75.5%                |
> | 5            | 72.2%                  | 72.9%                    | 75.0%                |
>
> BERTScore
>
> * To clarify, results in the current version of the manuscript indicate that **BERTScore *is* competitive with BLEU** in the Chatbot Arena analysis (Figure 1\) and in training experiments (Table 11, Appendix B.10). However, due to computational constraints, we did not pursue extensive training with BERTScore.
>
> BLEURT
>
> * This current manuscript does not mention BLEURT. We leave the exploration of more reference-based metrics like this to future work.
>
> ***`3. Concerns with Human Evaluation Reporting (line 320)`***
> Thank you for raising these concerns. Our reported soft preference rate was not intended to imply BLEUBERI’s superiority, but rather to reflect how often its outputs were **at least as good as** GRPO-RM’s (i.e., including ties and wins). This metric indicates general parity, not dominance. We acknowledge it could be misleading without context and will revise the manuscript to clearly distinguish it from strict preference counts.
>
> Regarding the Cohen’s Kappa of 0.34, we note that **our annotation task required subjective judgments on highly similar, high-quality responses to complex prompts from four benchmarks** (MT-Bench, ArenaHard v1/v2, WildBench). These often involved technical or domain-specific content, making clear preferences difficult. The frequent use of the “tie” option by one annotator further suggests the closeness in output quality. Prior studies have also observed that subjective evaluation tasks often yield lower inter-annotator agreement (Wang et al., 2024; Plank, 2022).
>
> Our human evaluation supports the key finding that **BLEUBERI-trained models can rival reward-model-trained counterparts in output quality.** We will revise the evaluation section to better highlight its limitations, clarify metric interpretation, and contextualize their relevance to our conclusions.
>
> References:
>
> * The “Problem” of Human Label Variation: On Ground Truth in Data, Modeling and Evaluation (Plank et al., 2022\)
> * HelpSteer2: Open-source dataset for training (Wang et al., 2024\)
>
> ## Questions
>
> ***`1. Interest in seeing the results on larger base model sizes (32B)`***
> We understand that results on a much larger base model would offer valuable insights into the effectiveness of simple reference-based reward functions like BLEU. However, due to limited compute resources, we leave such explorations to future work. We would like to highlight again that all training experiments (exploratory and final) cost about 1357 GPU hours (around **$4750 USD**), which represents a substantial computational cost for an academic lab.
>
> ***`2. More direct quantification of GRPO algorithm speedup between RM-8B and BLEU`***
> For two runs trained with BLEU and RM-8B as the reward, the average reward computation times over 250 steps (profiled by TRL) were:
>
> * RM-8B: \~1.15s
> * BLEU: \~0.03s
>
> Despite BLEU being much faster, both runs took roughly the same wall-clock time (\~2h45m). **In our setup, the main bottleneck was output generation per group**, which was slow due to using standard Hugging Face Transformers without fast inference backends like vLLM. As a result, BLEU’s speed advantage didn’t significantly impact end-to-end training time.
>
> However, in more optimized setups with faster generation or larger group sizes, BLEU’s efficiency could offer greater benefits. Not having to load a reward model should free up memory for larger batch sizes. In our experiments, we use identical hyperparameters for BLEUBERI and GRPO-RM for fairness. But in practice, **using lightweight metrics like BLEU could enable better scaling**.

---

> > ### Author Response · Authors · 2025-08-07
> > **Followup**
> >
> > Dear Reviewer dqb5,
> >
> > Thank you once again for your thoughtful review. If you have any follow-up questions or concerns regarding our rebuttal, we would be happy to provide additional clarification before the discussion period concludes.
> >
> > To summarize, we have included **new analysis results with ROUGE-L and ROUGE-L + BLEU ensemble**, and provided clarifications for other points you raised. In the next version of the manuscript, we also plan to incorporate **training results on larger models (32B)**, as per your suggestion.
> >
> > If our rebuttal has addressed your concerns, we would be grateful if you would consider adjusting your score accordingly. We sincerely appreciate your time and helpful feedback!

---

### Official Review · Reviewer_R3fD · 2025-07-04

**Clarity:** 4
**Significance:** 2
**Originality:** 3
**Rating:** 3
**Confidence:** 4

**Summary:**

This paper introduces BLEUBERI, an approach for aligning Large Language Models (LLMs) with human preferences by directly using BLEU, a simple string-matching metric, as a reward function in reinforcement learning (RL). Traditionally, LLM alignment relies on costly reward models that require extensive human-labelled preference data. The authors surprisingly demonstrate that BLEU effectively models human preferences for general instruction-following tasks, even rivalling large reward models in agreement. Specifically, BLEU with five synthetic references achieved a 74.2% agreement with human preferences, comparable to a 27B-parameter reward model's 75.6%.

**Questions:**

Please refer to weaknesses above.

1. How would this work be adapted to dealing with mathematical reasoning tasks? Any initial experiments or thoughts?
2. How does variability in reference quality impact the effectiveness of the training process
3. Can BLEU be prone to more subtle reward hacking?
4. Can this metric work well in multi-step reasoning tasks? Why was there a performance decline with enforced CoT reasoning? Could you elaborate please?

**Ethical Concerns:**

["NO or VERY MINOR ethics concerns only"]

**Final Justification:**

The "reference" is the part that I am not convinced about even after the final response by the authors. If you use existing LLMs or existing datasets, they have already put in a ton of effort/cost to generate that reference. I had requested for evaluation with not so great references to see if the performance can be attributed to the contribution of the paper. However, authors never answered or did experiments with regards to that.  So, I am sticking to my earlier scores.

**Limitations:**

Yes

**Quality:**

3

**Strengths And Weaknesses:**

Strengths

1. BLEU surprisingly matches strong reward models in modelling human preferences for general instruction following tasks
2. Cost effectiveness
3. Experimental evaluation demonstrating enhanced factuality and positive human evaluation

Weaknesses

1. Reliance on reference quality is the biggest concern
2. Absence of statistical significance results
3. Limited experiments on exploring effects of scaling model size, data volume or extensive training time
4. Known issues in mathematical reasoning tasks

---

> ### Author Rebuttal · Authors · 2025-07-31
>
> We appreciate the reviewer’s recognition of our work, including BLEUBERI’s competitive performance with GRPO-RM, BLEUBERI’s cost efficiency, and our strong results in both factuality and human evaluations. Below, we respond to each of the weaknesses and questions in detail.
>
> ## Weaknesses
>
> ***`1. Reliance on reference quality is the biggest concern`***
> We acknowledge this limitation, but **obtaining high-quality references is straightforward**. Such data can come from two sources:
>
> * **Synthetic outputs from frontier LLMs**: Generating synthetic data has become common in LLM training pipelines (Lambert et al., 2024; Qwen Team, 2024). These data can be highly effective in practice.
> * **Ground-truth responses from existing datasets**: Many publicly available datasets already provide high-quality reference answers. Prominent examples include Tulu 3 (Lambert et al., 2024), OpenOrca (Mukherjee et al., 2023), UltraChat (Ding et al., 2023).
>
> In contrast, **training reward models demands extensive human-labeled preference data, which is costly and time-consuming**. Lightweight reference-based metrics like BLEUBERI offer a more accessible and efficient path to alignment.
>
> References:
>
> * Tulu 3: Pushing Frontiers in Open Language Model Post-Training (Lambert et al., 2024\)
> * Qwen2.5 Technical Report (Qwen Team, 2024\)
> * Enhancing Chat Language Models by Scaling High-quality Instructional Conversations (Ding et al., 2023\)
> * Orca: Progressive Learning from Complex Explanation Traces of GPT-4 (Mukherjee et al., 2023\)
>
> ***`2. Absence of statistical significance results`***
> We thank the reviewer for raising this point. In response, **we reran benchmark evaluations with bootstrapping and report results below**. Overall, we find that:
>
> * Both **BLEUBERI and GRPO-RM show consistent improvements over the base model across all three model types**, as suggested by minimal or no overlap between their confidence intervals and those of the base model.
> * Between BLEUBERI and GRPO-RM, the confidence intervals almost always overlap, suggesting that **BLEUBERI performs on par with GRPO-RM overall**.
>
> We will add these results into the next version of the manuscript.
>
> **Benchmark results with bootstrapping:**
> | Base model       | Variant   | MT-Bench | 95% CI         | ArenaHard v1 | 95% CI         | ArenaHard v2 | 95% CI         | WildBench | 95% CI         |
> |------------------|-----------|----------|----------------|--------------|----------------|--------------|----------------|-----------|----------------|
> | Qwen2.5-7B       | Base      | 63.5     | (-4.0, \+4.1)   | 16.2         | (-1.6, 1.6)     | 5.6          | (-0.8, \+0.7)    | 51.8      | (-1.6, \+1.6)   |
> |                  | SFT       | 67.5     | (-4.3, \+3.7)   | 22.1         | (-2.1, 2.2)     | 9.9          | (-1.0, \+1.1)    | 60.5      | (-1.3, \+1.3)   |
> |                  | GRPO-RM   | 76.9     | (-3.8, \+3.2)   | 29.8         | (-2.2, 2.2)     | 12.2         | (-1.2, \+1.3)    | 64.8      | (-1.1, \+1.3)   |
> |                  | BLEUBERI  | 70.8     | (-3.8, \+3.6)   | 29.3         | (-1.8, 2.7)     | 12.9         | (-1.4, \+1.2)    | 65.4      | (-1.3, \+1.3)   |
> |                  | Instruct  | 78.8     | (-3.5, \+3.3)   | 37.9         | (-2.8, 2.5)     | 16.8         | (-1.1, \+1.5)    | 71.0      | (-1.2, \+1.1)   |
> | Qwen2.5-3B       | Base      | 61.1     | (-3.9, \+3.7)   | 7.0          | (-1.0, 1.0)     | 3.1          | (-0.6, \+1.0)    | 49.0      | (-1.3, \+1.5)   |
> |                  | SFT       | 59.6     | (-4.2, \+4.2)   | 9.8          | (-1.1, 1.6)     | 4.0          | (-0.6, \+0.8)    | 55.6      | (-1.4, \+1.3)   |
> |                  | GRPO-RM   | 67.8     | (-4.2, \+4.0)   | 12.8         | (-1.3, 1.6)     | 5.1          | (-0.6, \+0.6)    | 59.2      | (-1.2, \+1.3)   |
> |                  | BLEUBERI  | 64.6     | (-4.1, \+3.9)   | 11.0         | (-1.3, 1.3)     | 3.7          | (-0.5, \+0.6)    | 56.1      | (-1.2, \+1.3)   |
> |                  | Instruct  | 70.0     | (-4.2, \+3.8)   | 18.9         | (-2.0, 1.9)     | 6.7          | (-0.8, \+0.9)    | 63.0      | (-1.2, \+1.3)   |
> | Llama-3.1-8B     | Base      | 52.9     | (-4.3, \+4.2)   | 6.8          | (-1.0, 1.2)     | 2.1          | (-0.5, \+0.6)    | 55.0      | (-1.3, \+1.3)   |
> | (SFT init.)      | SFT       | 56.7     | (-4.5, \+4.5)   | 12.6         | (-1.6, 1.7)     | 3.0          | (-0.6, \+1.0)    | 60.3      | (-1.1, \+1.3)   |
> |                  | GRPO-RM   | 57.2     | (-4.4, \+4.3)   | 9.7          | (-1.2, 1.4)     | 1.6          | (-0.4, \+0.6)    | 57.8      | (-1.2, \+1.4)   |
> |                  | BLEUBERI  | 56.9     | (-4.4, \+4.3)   | 10.1         | (-1.4, 1.5)     | 2.3          | (-0.4, \+0.6)    | 59.5      | (-1.2, \+1.1)   |
> |                  | Instruct  | 65.8     | (-3.8, \+3.6)   | 24.9         | (-2.4, 2.8)     | 5.8          | (-0.8, \+1.1)    | 64.2      | (-1.2, \+1.1)   |
>
> Arena-Hard v1/v2 already include bootstrapping in their official code: they draw 100 resamples and re-fit a Bradley-Terry model for each, which is computationally intensive, hence the cap at 100\. We reran the scripts and report the median point estimates and 95% confidence intervals (CIs). For MT-Bench and WildBench, we implemented a similar procedure with 1000 full-size prompt resamples per model, reporting the median score and two-sided 95% CI.
>
> ***`3. Limited experiments on exploring effects of scaling model size, data volume or extensive training time`***
> In our preliminary experiments, we trained for multiple epochs but observed no significant performance gains for either BLEUBERI or GRPO-RM. One possible explanation is that larger models might benefit more from longer training. Regarding data volume, we note that our setup aligns with that of the Qwen3 pipeline (Qwen Team, 2025), which trains exclusively on 4,000 challenging samples, as noted on Line 204\.
>
> Overall, we agree that more scaling experiments can provide helpful insights. However, due to limited compute resources, we leave such explorations to future work. We would like to highlight again that all training experiments (exploratory and final) cost about 1357 GPU hours (around **$4750 USD**), which represents a substantial computational cost for an academic lab.
>
> References:
>
> * Qwen3 Technical Report (Qwen Team, 2025\)
>
> ***`4. Known issues in mathematical reasoning tasks`***
> We assume the reviewer is referring to the results shown in **Figure 7**, which shows that on the Math/Reasoning domain, the gap between BLEU’s and the two reward models’ human agreement rates is large. We would like to reiterate that **the primary focus of this work is on general-domain instruction following**. For structured domains like math, it is straightforward to develop reward functions that checks the correctness of the final answer. If model outputs consisted solely of final answers, BLEU would essentially function as a correctness-based reward. However, many Math/Reasoning examples in the Chatbot Arena dataset include intermediate reasoning steps, which reduce BLEU’s effectiveness as a proxy for correctness. In our experiments, we chose not to optimize for any specific domain.
>
> ## Questions
>
> ***`1. How would this work be adapted to dealing with mathematical reasoning tasks? Any initial experiments or thoughts?`***
> Building on the discussion of Weakness 4, one potential way to improve both general-domain and math performance is to combine BLEU and a correctness reward into a single objective function, where BLEU could be used to assess the quality of reasoning traces. We plan to explore this approach in a future version of the manuscript.
>
> ***`2. How does variability in reference quality impact the effectiveness of the training process`***
> We provide training results with different references in **Table 9** in the Appendix. These results generally align with our findings from Section 2.2 (Line 124), suggesting that stronger reference models will lead to better results, which is intuitive. As discussed under Weakness 1, obtaining high-quality references is straightforward.
>
> ***`3. Can BLEU be prone to more subtle reward hacking?`***
> We interpret the reviewer’s comment as referring to subtle, hard-to-detect artifacts. As noted in **Section 4.1 (Lines 280–294)**, GRPO-RM models often begin with “Sure\!”, while BLEUBERI models favor “Certainly\!”—though this trait is not unique to BLEUBERI. Importantly, human evaluations (Section 4.3) rate BLEUBERI and GRPO-RM responses as equally preferable, and benchmark results (Table 1\) and factuality assessments (Section 4.2) show that **any such artifacts do not meaningfully impact perceived quality**.
>
> ***`4. Can this metric work well in multi-step reasoning tasks? Why was there a performance decline with enforced CoT reasoning? Could you elaborate please?`***
> **Prior work shows that explicit reasoning does not always help**, especially when instruction adherence is the main goal. In our experiments, enforcing CoT reasoning reduced performance, consistent with Li et al. (2025), who found that CoT can distract from constraint-relevant tokens and introduce errors. Sprague et al. (2024) similarly observed that **CoT primarily benefits math and symbolic tasks, with little or no gain and sometimes harm on general instruction-following**. Our BLEUBERI results align with these findings, supporting the idea that reasoning should be applied selectively.
>
> It is possible that larger models or longer generation lengths may allow more effective use of reasoning in general instruction-following tasks. We plan to explore this in a future version of the manuscript.
>
> References:
>
> * When Thinking Fails: The Pitfalls of Reasoning for Instruction-Following in LLMs (Li et al., 2025\)
> * To CoT or not to CoT? Chain-of-thought helps mainly on math and symbolic reasoning (Sprague et al., 2024\)

---

> > ### Comment · Reviewer_R3fD · 2025-08-07
> >
> > Thanks for the detailed results and taking time to run new experiments.
> >
> > The "reference" is the part that I am not convinced about still. If you use existing LLMs or existing datasets, they have already put in a ton of effort/cost to generate that reference. Aren't you piggy backing on those efforts? Have you had a chance to try cheap references ? In fact, what would be cheap references.

---

> > > ### Author Response · Authors · 2025-08-07
> > >
> > > Thank you for the followup questions! While it's true that training frontier LLMs takes a lot of effort and money, **using synthetic data generated by these models to improve smaller models has become a widely adopted and cost-effective methodology**. Several seminal works have demonstrated the effectiveness of this approach across both academia and industry.
> > >
> > > - Stanford Alpaca [1] showed that a small 7B LLaMA model could achieve instruction-following performance comparable to much larger models by fine-tuning on 52K examples generated by OpenAI’s text-davinci-003, costing **only ~$600**. The results validated the **effectiveness of using frontier LLMs for low-cost instruction tuning**, establishing a new paradigm in the field.
> > >
> > > - WizardLM [2] prompted a strong LLM to iteratively rewrite a base set of simple instructions into more complex tasks. A LLaMA-based model fine-tuned on this evolved dataset demonstrated capabilities comparable to ChatGPT on many tasks, showing that **LLMs can generate increasingly sophisticated training data beyond what humans typically author**.
> > >
> > > - [3] explored instruction tuning using GPT-4 as the synthetic data generator. They showed that a small LLaMA model fine-tuned on GPT-4–generated instruction-following QA pairs achieved impressive performance, highlighting the **benefits of using state-of-the-art teacher models to supervise and bootstrap smaller students**.
> > >
> > > - Persona Hub [4] shows that prompting LLMs with diverse synthetic personas can generate high-quality training data at massive scale. A 7B model fine-tuned on 1M persona-generated math problems achieved 64.9\% on MATH, rivaling GPT-4-turbo-preview, demonstrating that **frontier models can be used to extract capabilities into smaller models via synthetic data**.
> > >
> > > The use of synthetic data has become common in many modern LLM training pipelines, including Llama [5], Qwen [6], OLMo [7], etc. **Compared to collecting human-authored references, generating synthetic data is already significantly more scalable and cost-effective.** Our method provides an alternative way to use such synthetic references that has advantages compared to SFT / training with reward models in terms of performance and efficiency.
> > >
> > > References:
> > > - [1] Stanford Alpaca: An Instruction-following LLaMA model (Taori et al., 2023)
> > > - [2] WizardLM: Empowering large pre-trained language models to follow complex instructions (Xu et al., 2023)
> > > - [3] Instruction Tuning with GPT-4 (Peng et al., 2023)
> > > - [4] Scaling Synthetic Data Creation with 1,000,000,000 Personas (Ge et al., 2024)
> > > - [5] The Llama 3 Herd of Models (Llama Team, 2024)
> > > - [6] Qwen3 Technical Report (Qwen Team, 2025)
> > > - [7] 2 OLMo 2 Furious (OLMo Team, 2024)

---

> ### Author Response · Authors · 2025-08-07
> **Followup**
>
> Dear Reviewer R3fD,
>
> Thank you once again for your thoughtful review. If you have any follow-up questions or concerns regarding our rebuttal, we would be happy to provide additional clarification before the discussion period concludes.
>
> To summarize, we have included **new benchmark evaluation results** with bootstrapping and clarified several of the points you raised. In the next version of the manuscript, we also plan to incorporate **training results on larger models and more data**, as per your suggestion.
>
> If our rebuttal has addressed your concerns, we would be grateful if you would consider adjusting your score accordingly. We sincerely appreciate your time and helpful feedback!

---

### Decision · Program_Chairs · 2025-09-17

**Decision:**

Accept (poster)

**Comment:**

This paper introduces BLEUBERI, a method that leverages BLEU as a reward signal in GRPO for instruction-following alignment, replacing the need for costly reward models. Despite BLEU being a relatively simple n-gram matching metric, the paper shows that it can be surprisingly effective, achieving performance comparable to GRPO-RM across most settings.

The reviewers agreed that the core idea is both practical and novel. They highlighted several strengths: BLEU is efficient to compute compared to expensive reward models, and BLEUBERI achieves competitive or even superior results across four challenging benchmarks and three different base models.  The paper also shows that BLEUBERI-trained models tend to be more factually grounded than those trained with reward models or SFT, and the authors provide thoughtful analyses (agreement study, synthetic data experiments, metric comparisons, and qualitative analyses) that strengthen the work. While the evaluation could be further expanded, the results presented, along with the additional rebuttal experiments, provide convincing evidence of the method’s potential. In the discussion, two reviewers expressed strong enthusiasm for the paper, and those who participated were overall positive, including one of the weak rejects. The other weak reject did not take part in the discussion.

Overall, the paper makes a useful and timely contribution. This is a simple but elegant idea, executed well, and the results are both interesting and insightful. The work provides a practical and cost-effective alternative to reward-model-based alignment that will be useful for the community.  I recommend acceptance, and I encourage the authors to incorporate the reviewers’ feedback and the new rebuttal experiments into the final version to maximize the paper’s impact.